# CURATED LLM: SYNERGY OF LLMS AND DATA CURATION FOR TABULAR AUGMENTATION IN ULTRA LOW-DATA REGIMES

## ABSTRACT

Machine Learning (ML) in low-data settings remains an underappreciated yet crucial problem. This challenge is pronounced in low-to-middle income countries where access to large datasets is often limited or even absent. Hence, data augmentation methods to increase the sample size of datasets needed for ML are key to unlocking the transformative potential of ML in data-deprived regions and domains. Unfortunately, the limited training set constrains traditional tabular synthetic data generators in their ability to generate a large and diverse augmented dataset needed for ML tasks. To address this technical challenge, we introduce CLLM, which leverages the prior knowledge of Large Language Models (LLMs) for data augmentation in the low-data regime. While diverse, not all the data generated by LLMs will help increase utility for a downstream task, as for any generative model. Consequently, we introduce a principled curation process, leveraging learning dynamics, coupled with confidence and uncertainty metrics, to obtain a high-quality dataset. Empirically, on multiple real-world datasets, we demonstrate the superior performance of LLMs in the low-data regime compared to conventional generators. We further show our curation mechanism improves the downstream performance for all generators, including LLMs. Additionally, we provide insights and understanding into the LLM generation and curation mechanism, shedding light on the features that enable them to output high-quality augmented datasets. CLLM paves the way for wider usage of ML in data scarce domains and regions, by allying the strengths of LLMs with a robust data-centric approach.

## 1 INTRODUCTION

**No data, No Machine Learning.** Machine learning (ML) has transformed numerous industries, but its wider adoption is hindered by a pervasive roadblock: insufficient data. Specifically, the use of ML algorithms presumes the availability and access to large datasets for training, be it in the form of labeled or unlabeled data. Unfortunately, real-world domains are often data scarce: (i) in healthcare and finance, collecting annotations can be expensive or practically impossible; (ii) in developing and low-to-middle income countries (LMICs), digital infrastructure (such as electronic healthcare records (EHRs)) can be limited or nonexistent (Ade-Ibijola & Okonkwo, 2023; Asiedu et al., 2023; Owoyemi et al., 2020; Mollura et al., 2020; Alami et al., 2020; Ciecierski-Holmes et al., 2022) and (iii) within large datasets, there can be (ethnic) minorities that are underrepresented. This lack of data has serious consequences: to sideline these settings to the peripheries of ML advancements and prevent the development of accurate models. How can we build a reliable ML model in this *low-data regime*, where we have so few samples? Solving this problem is a major opportunity that would unlock the potential of ML across society, domains, and regions.

**Aim.** To address this important yet undervalued low-data problem, we aim to augment the *small labeled dataset* ($n < 100$) with synthetic samples. We focus on tabular data, as defining augmentations is non-trivial and can easily result in nonsensical or invalid samples. Moreover, tabular domains like healthcare (of value in LMICs) are often where data scarcity is acute.

**Related work.** Data augmentation is a widely used and different approach to address data scarcity in tabular data contexts. Methods are either based on generative models (Ghosheh et al., 2023; Biswas

Figure 1: CLLM uses a small dataset $D_{\text{train}}$ and a frozen black-box LLM to generate a larger synthetic set $D_{\text{syn}}$. The curator computes the learning dynamics of samples in $D_{\text{syn}}$, assessing samples based on their aleatoric uncertainty and predictive confidence, then curates $D_{\text{syn}}$ with the goal that a downstream model trained on the curated $D_{\text{curated}}$ will have improved performance.

et al., 2023; Wang & Pai, 2023; Machado et al., 2022; Tanaka & Aranha, 2019) such as GANs (Xu et al., 2019), VAEs (Xu et al., 2019), Normalizing Flows (Papamakarios et al., 2021), Score-based models (Kotelnikov et al., 2022; Kim et al., 2022), or alternatively traditional methods such as SMOTE (Chawla et al., 2002; Wang & Pai, 2023; Machado et al., 2022). However, in ultra low-data regimes ($n < 100$), the training data may not describe the full data distribution well, despite it being i.i.d. draws. Consequently, this harms conventional methods since the augmented data may not be sufficiently diverse and accurate, restricting the generalizability of predictive models trained on such data. Tangentially, prior works have tackled data scarcity in the tabular setting via the lens of transfer learning, where prior knowledge can be transferred from a pretrained model (Levin et al., 2022; Jin & Ucar, 2023) or a knowledge graph (Margeloiu et al., 2022; Ruiz et al., 2023), which might not be available in all settings. Recent work has shown the potential of fine-tuning Large Language Models (LLMs) for tabular data generation (Borisov et al., 2023). While LLMs offer some degree of prior knowledge, there are two challenges in our setting. First, it is computationally expensive to fine-tune LLMs, while needing specialized hardware —luxuries often not available in LMICs, thereby limiting applicability in such settings. Second, fine-tuning often assumes a large number of samples. In our low-data setting it could lead to overfitting and low-quality generated samples, and hence poor downstream models—as we show for Borisov et al. (2023) in Sec. 3.

**Curated LLMs.** To address these challenges, we propose Curated LLM (CLLM). First, CLLM leverages the in-context capabilities of LLMs for generation, thereby reducing the computational burden. We also posit for the low-data regime; the diverse pretraining corpus of LLMs carries valuable prior knowledge, which may offer more diversity in their generation compared to other conventional tabular generators. Of course, LLMs are not perfect. Balancing the utility of LLMs against the risk of noisy, irrelevant data is important for downstream performance, hence requiring systemic assessment of the generated data. In fact, this issue is vital for *any* generative model.

This motivates the second key aspect of CLLM, i.e. post-generation data curation. This addresses the *overlooked* aspect that not all of the synthetic samples are useful to downstream model performance, with some samples even harmful. We anchor our approach with ideas from learning theory that show the behavior of individual data samples during training, called learning dynamics, provides a salient signal about the value of samples to a learner (Arpit et al., 2017; Arora et al., 2019; Li et al., 2020). To provide intuition, samples with variable predictions might be considered ambiguous or other samples might never be learned correctly and could harm a model. In CLLM, we study the learning dynamics of the synthetic data samples, with respect to a model trained on the small real dataset. We then analyze these dynamics by computing two key metrics: confidence and aleatoric (data) uncertainty. These metrics form the basis for curating the synthetic samples. We aim to enable a highly performant downstream model when trained on the curated dataset.

> **Contributions:** CLLM is a novel data augmentation approach allying the strengths of LLMs with a robust data curation mechanism to improve data augmentation in the *ultra low-data regime* ($n < 100$), bringing several contributions: ① **Improved performance:** we empirically demonstrate on 7 real-world datasets that CLLM enables superior downstream performance compared to 6 widely used tabular data generative models and data augmentation techniques. ② **Value of curation:** we show the *overlooked* aspect of synthetic data curation improves downstream performance across the generative models. This highlights the flexibility and broad utility of our curation mechanism for data augmentation. ③ **Insights:** we dissect the two aspects of CLLM (LLM and data curation) along a variety of dimensions, providing insights and understanding into why the approach is beneficial. We show the largest gains are for underrepresented subgroups and in ultra low-data settings. These contributions pave the way towards wider usage of ML across society, domains and regions.

**Ethical considerations.** LLMs may make errors and may reflect or exacerbate societal biases that are present in their data (Li et al., 2023). Though the curation in CLLM improves synthetic data quality, it does not directly aim to remove biases. The quality and fairness of generated data should always be evaluated. More research into LLM bias is required before methods like CLLM should be applied to real-world sensitive settings like healthcare and finance.

## 2    CLLM: SYNERGY OF LLM GENERATION AND DATA CURATION

**Set-up.** Given feature space $\mathcal{X}$, and label space $\mathcal{Y} = \{1, ..., k\}$, we assume that we only have a small labeled dataset $D_{\text{train}} = \{(x_i, y_i)\}_{i=1}^n$, with $x_i \in \mathcal{X}$, $y_i \in \mathcal{Y}$ and $n < 100$ (ultra-low data setting). Assume $D_{\text{train}}$ is drawn i.i.d. from the real distribution $p_R(X, Y)$. We also assume access to a pretrained LLM to generate samples. We denote the output distribution of the LLM as $p_\Phi(X, Y)$, with $\Phi$ containing parameters that we control (e.g., input prompts). Our goal is to generate a dataset to augment the small $D_{\text{train}}$, and subsequently use it to train a classifier $f : \mathcal{X} \to \mathcal{Y}$. Successful augmentation will provide a better classifier $f$, than if we had trained $f$ on the small $D_{\text{train}}$ itself. We measure downstream performance on a separate held-out dataset of real data, $D_{\text{test}}$.

**Our Approach.** To address this challenge, we introduce CLLM, an approach for data augmentation in low-data regimes. As shown in Figure 1, CLLM leverages LLMs to **generate** a synthetic dataset $D_{\text{syn}}$ using a small dataset $D_{\text{train}}$ (Sec. 2.1). It exploits the LLMs' prior knowledge via in-context learning (ICL) and contextual information. CLLM then **curates** $D_{\text{syn}}$ by analyzing the learning dynamics of samples in $D_{\text{syn}}$ based on predictive confidence and aleatoric (data) uncertainty. These metrics are obtained by training a supervised model on $D_{\text{train}}$. We leverage them to define a curated dataset $D_{\text{curated}}$, which is used to train a downstream classifier (Sec. 2.2).

In each sub-section we describe and motivate the design of the different aspects of CLLM (LLM and curation). Furthermore, we provide insights and understanding into their role in improving data utility, which we later quantify on multiple real-world datasets in Sec. 3.

### 2.1    DATA GENERATION WITH LLMS BASED ON A SMALL $D_{\text{train}}$

As outlined in Sec. 1, in the ultra low-data regime, conventional tabular generative models (e.g. CTGAN, TVAE) are constrained by the limited $D_{\text{train}}$ and may not generate sufficiently diverse and/or accurate synthetic data. To address this challenge, we propose to leverage LLMs, building on their large-scale pretraining. We first outline the desirable features of LLMs for tabular data generation when we have very few samples, then describe design choices to satisfy these.

- **Prior knowledge.** LLMs have been pretrained with a vast corpus of information (Chowdhery et al., 2022; Singhal et al., 2023). When prompted to generate samples with limited real data, LLMs can leverage this encoded prior information about similar problems and feature-label relationships to enhance both accuracy and diversity of generation.
- **Contextual understanding.** LLMs can process background and contextual information about the problem via natural language (Yang et al., 2023). For example, a high-level description of the task, features and their meanings can be conveniently described through natural language. Such information is unavailable to conventional generators that only utilize numerical examples.
- **Few-shot capabilities.** LLMs have demonstrated proficiency in generalizing to tasks with just a few examples (Brown et al., 2020; Wei et al., 2023; Mirchandani et al., 2023). In the context of generation, we envision the idea of in-context generation using limited real examples.

To benefit from these capabilities, we craft the LLM prompt with three different parts (see Fig. 1): (1) *Background*: text description of the dataset and task (e.g. predict Covid mortality). Additionally, we include a description of what each feature means, explicitly prompting the LLM to use prior knowledge about these features. (2) *Examples*: we serialize the samples in $D_{\text{train}}$ as example demonstrations and provide both the features and the label in text format. (3) *Instructions*: To generate a synthetic dataset $D_{\text{syn}}$, we instruct the LLM to leverage the contextual information and provided examples as an i.i.d. draw from the distribution. We instruct the LLM to identify structural and feature-label relationships in the data and generate diverse data following the structure and format of the provided examples. We provide more details on the prompts in Appendix B.

**Motivation for a frozen LLM.** Using a frozen black-box LLM (e.g. GPT-4 or GPT-3.5) is computationally cheaper and requires less specialized hardware (i.e. GPUs) compared to fine-tuning. This relates to settings described in Sec. 1, such as LMICs, where we may not have the computational

resources to fine-tune an LLM. Even in settings where fine-tuning is possible, we show empirically in Sec. 3 that LLM fine-tuning (e.g. GReaT baseline) is suboptimal in ultra-low data settings ($n < 100$) compared to providing in-context examples coupled with curation.

> **Dissecting the LLM's generative features.** We now investigate various dimensions to understand and illustrate empirically the appealing features of LLMs as data generators in the low-data regime, and how our design choices unlock them. We take the Brazilian *Covid-19* dataset (Baqui et al., 2020) as a running example and focus on GPT-4 as the LLM.

▶ **GPT-4 extrapolates to unseen regions of the manifold.** We compare the samples generated by GPT-4 to TVAE, a widely used tabular data generator. We consider $D_{\text{oracle}}$, a held-out dataset from the same distribution as $D_{\text{train}}$, such that $|D_{\text{oracle}}| \gg |D_{\text{train}}|$, thereby providing an approximation for the true manifold. The t-SNE plots in Fig. 2 shows, when $D_{\text{train}}$ is very small ($n = 20$ samples), that its samples do not cover all regions of $D_{\text{oracle}}$. For example, $D_{\text{train}}$ does not contain samples from specific demographic subgroups (e.g. people with age 40 or below). As expected, TVAE only generates samples constrained by the limited $D_{\text{train}}$. In contrast, GPT-4 is capable of extrapolating and generating samples even in unseen regions of $D_{\text{train}}$, thereby better covering $D_{\text{oracle}}$. This stems from its *contextual understanding* of the features, unlocking the use of its *prior knowledge*. It leads to better coverage in the low-data regime, consequently aiding in superior downstream performance, as we show in Table 3. As $n$ increases ($\geq 100$), $D_{\text{train}}$ provides better coverage, which naturally benefits both GPT-4 and TVAE. This result shows how prior knowledge encoded in LLMs addresses shortcomings of conventional generative approaches (e.g. TVAE) in the low-data regime.

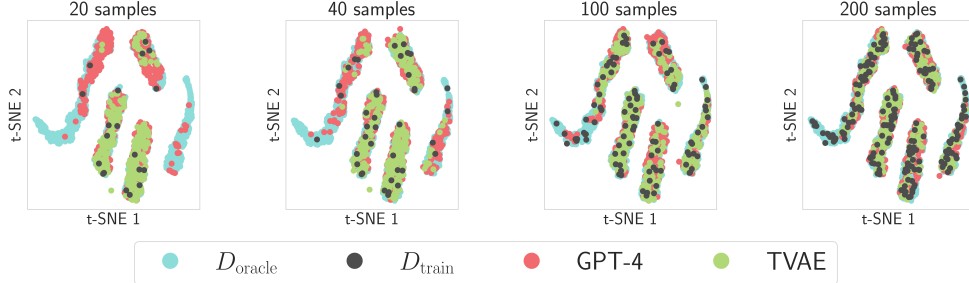

Figure 2: GPT-4 is able to extrapolate to regions of the oracle (true manifold) even where there is no training data covering them, as can be seen by the overlap with the turquoise dots, with the effect more pronounced when $D_{\text{train}}$ is small

▶ **GPT-4 benefits underrepresented groups the most.** Having illustrated the extrapolation capabilities of GPT-4, we now ask: *where does augmentation benefit downstream performance the most?* We evaluate performance gains for different demographic subgroups, such as age groups and ethnic groups (Amarela, Prada). Fig. 3 shows the performance gain obtained by training a classifier on data generated by GPT-4 compared to training on the small $D_{\text{train}}$. The greatest gains, on average, are for subgroups for which we have *no data* in $D_{\text{train}}$, yet GPT-4 can extrapolate and generate samples for these subgroups. This further validates the rationale of extrapolation via prior knowledge being a key source of gain for GPT-4. Table 1 shows fine-grained results (across 10 different seeds) for the 5 subgroups that benefit the most from data augmentation, which are small-sized demographic subgroups. This finding has real-world implications for equity, as it shows we can improve performance for underrepresented subgroups even when we lack data or collecting data is difficult or costly.

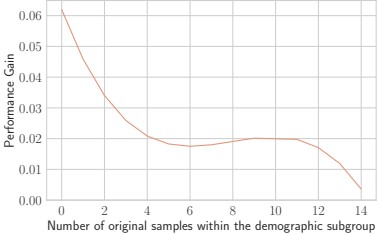

Figure 3: Subgroups with fewest samples in $D_{\text{train}}$ benefit the most from data augmentation, on average.

Table 1: Deep dive into the top 5 demographic subgroups in the Covid dataset with the largest gains, across 10 seeds, for $|D_{\text{train}}| = 20$. GPT-4 improves performance on the smallest groups.

| Subgroup | $n_{\text{samples}}$ in $D_{\text{train}}$ (min - max) | Avg. Acc. Gain v. $D_{\text{train}}$ GPT-4 | TVAE |
|---|---|---|---|
| Age_40 | 0-6 | **6.38** +- 2.09 | -3.37 +- 2.86 |
| Liver | 0-1 | **3.85** +- 3.37 | -13.1 +- 3.38 |
| Renal | 0-3 | **4.52** +- 2.01 | -18.0 +- 3.22 |
| Amarela | 0-1 | **8.71** +- 1.40 | -2.03 +- 2.88 |
| Parda | 3-11 | **5.07** +- 1.50 | -6.57 +- 1.61 |

▶ **Importance of contextual information in the prompt.** A natural question is: *how important is the prompt to elicit the prior knowledge of the LLM?* We explore two variants: (1) *Prompt w/ context*: provides contextual information including background about the dataset, feature names and descriptions (our approach) and (2) *Prompt w/ no context*: only provides the numerical in-context examples (ablation). Fig. 4 qualitatively shows that not including contextual knowledge in the prompt gives lower coverage of $D_{\text{oracle}}$ with less extrapolation beyond $D_{\text{train}}$. We quantify this in Table 2 using Precision (Quality) and Recall (Diversity) metrics (Sajjadi et al., 2018), as well as Utility (Downstream performance). GPT-4 *with contextual information* has superior precision and recall in the ultra-low data setting. Furthermore, we show that *the lack* of contextual information in the prompt significantly harms the precision (quality) of the data even compared to TVAE. This highlights that LLMs need guidance, as we are only able to get the extrapolation and performance benefits by including contextual information, further motivating our design choices in the prompt.

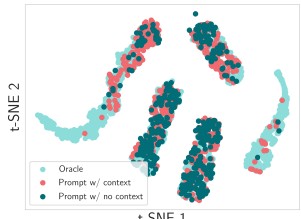

Figure 4: Contextual information in the prompt is important for extrapolation.

Table 2: Including contextual information in the prompt improves precision (P), recall (R), and utility (U) in low-sample settings (results shown for the Covid dataset).

| $n_{\text{samples}}$ in $D_{\text{train}}$ | GPT-4 w/ context | | | GPT-4 no context | | | TVAE | | |
|---|---|---|---|---|---|---|---|---|---|
| | P | R | U | P | R | U | P | R | U |
| 20 | **0.41**(0.04) | **0.87**(0.03) | **0.74**(0.01) | 0.13(0.0) | 0.82(0.01) | 0.66(0.01) | 0.33(0.07) | 0.50(0.03) | 0.59(0.02) |
| 40 | **0.40**(0.01) | **0.91**(0.01) | **0.76**(0.0) | 0.11(0.0) | 0.89(0.0) | 0.69(0.0) | 0.27(0.01) | 0.68(0.01) | 0.62(0.03) |
| 100 | **0.42**(0.01) | 0.86(0.02) | **0.75**(0.01) | 0.11(0.01) | **0.90**(0.01) | 0.74(0.01) | 0.39(0.02) | 0.67(0.03) | 0.64(0.06) |
| 200 | 0.44(0.02) | 0.85(0.02) | **0.75**(0.0) | 0.08(0.01) | **0.90**(0.0) | 0.60(0.01) | **0.47**(0.0) | 0.73(0.01) | 0.65(0.02) |

## 2.2 DATA CURATION WITH LEARNING DYNAMICS

When prompted with $\Phi$ (which contains the in-context samples of $D_{\text{train}}$), the LLM generates samples from a distribution $p_\Phi(X, Y)$ that approximates $p_R(X, Y)$, implicitly exploiting its large-scale pretraining and few-shot capabilities. LLMs are of course not perfect and could generate noisy samples, hence this distribution may be inaccurate [1]. To make this distribution more relevant to the downstream task, we include a data curation mechanism. Specifically, we focus on the noisy feature-label relationship $p_\Phi(Y|X)$, for which we expect $p_\Phi(Y|X) \neq p_R(Y|X)$ given the small size of $D_{\text{train}}$. This motivates us to curate $D_{\text{syn}}$ and discard likely mislabeled samples.

We anchor our approach with ideas from learning theory that show the behavior of individual samples during model training (called *learning dynamics*) contains signal about the nature of the samples themselves (Arpit et al., 2017; Arora et al., 2019; Li et al., 2020). Some samples are easily and confidently predicted over different model checkpoints, whereas other samples might be challenging (e.g. due to mislabeling) and hence might be incorrectly predicted for the given label. Consequently, we operationalize *learning dynamics* as the basis of our curation mechanism. Specifically, we analyze samples in $D_{\text{syn}}$ by studying their learning dynamics computed with a classifier trained on $D_{\text{train}}$. We then categorize and filter samples in $D_{\text{syn}}$, and produce a curated dataset $D_{\text{curated}} \subset D_{\text{syn}}$.

**Learning dynamics.** We now formalize how we compute learning dynamics for individual samples. Assume that a classifier $f$ is trained in an iterative scheme (e.g. neural networks or XGBoost trained over iterations) on $D_{\text{train}}$, which makes it possible to analyze the learning dynamics of samples in $D_{\text{syn}}$ over these iterations. The classifier $f$ should be at least as flexible as the model that the practitioner intends to use for the downstream task. $f$ is trained from scratch on $D_{\text{train}}$ and goes through $e \in [E]$ different checkpoints leading to the set $\mathcal{F} = \{f_1, f_2, \ldots, f_E\}$, such that $f_e$ is the classifier at the $e$-th checkpoint. Let $[f_e(x)]_y$ denote the predicted probability for class $y$ and sample $x$. Our goal is to assess the learning dynamics of samples in $D_{\text{syn}}$ over these $E$ training checkpoints, while we train $f$ on $D_{\text{train}}$. For this, we define $H$, a random variable following a uniform distribution $\mathcal{U}_\mathcal{F}$ over the set of checkpoints $\mathcal{F}$. Specifically, given $H = h$ and a sample $(x, y)$, we define the correctness in the prediction of $H$ as a binary random variable $\hat{Y}_\mathcal{F}(x, y)$ with the following conditional:

$$P(\hat{Y}_\mathcal{F}(x, y) = 1 | H = h) = [h(x)]_y \text{ and } P(\hat{Y}_\mathcal{F}(x, y) = 0 | H = h) = 1 - P(\hat{Y}_\mathcal{F}(x, y) = 1 | H = h).$$

---

[1] We could finetune the model on the scarce $D_{\text{train}}$ we have, but is likely to still lead to overfitting due to the extreme data scarcity and LLM parameter size.

**Curation metrics.** Equipped with a probabilistic interpretation of the predictions of a model, we now define two characterization metrics that we use for curation: (i) average confidence and (ii) aleatoric (data) uncertainty, inspired by (Kwon et al., 2020; Seedat et al., 2022a).

**Definition 2.1** (Average confidence). For any set of checkpoints $\mathcal{F} = \{f_1, ..., f_E\}$, the average confidence for a sample $(x, y)$ is defined as the following marginal:

$$\bar{P}_{\mathcal{F}}(x, y) := P(\hat{Y}_{\mathcal{F}}(x, y) = 1) = \mathbb{E}_{H \sim \mathcal{U}_{\mathcal{F}}}[P(\hat{Y}_{\mathcal{F}}(x, y) = 1 | H)] = \frac{1}{E} \sum_{e=1}^{E} [f_e(x)]_y \qquad (1)$$

**Definition 2.2** (Aleatoric uncertainty). For any set of checkpoints $\mathcal{F} = \{f_1, ..., f_E\}$, the aleatoric uncertainty for a sample $(x, y)$ is defined as:

$$v_{al,\mathcal{F}}(x, y) := \mathbb{E}_{H \sim \mathcal{U}_{\mathcal{F}}}[Var(\hat{Y}_{\mathcal{F}}(x, y) | H)] = \frac{1}{E} \sum_{e=1}^{E} [f_e(x)]_y (1 - [f_e(x)]_y) \qquad (2)$$

Intuitively, for binary classification ($k = 2$), the aleatoric uncertainty for a sample $x$ is maximized when $[f_e(x)]_y = \frac{1}{2}$ for all checkpoints $f_e$, akin to random guessing. Recall aleatoric uncertainty captures the inherent data uncertainty, hence is a principled way to capture issues such as mislabeling. This contrasts epistemic uncertainty, which is model-dependent and can be reduced simply by increasing model parameterization (Hüllermeier & Waegeman, 2021).

Having defined sample-wise confidence and aleatoric uncertainty, we characterize samples in $D_{\text{syn}}$ into two categories, namely *Selected* and *Discarded*. Given a sample $(x, y)$, a set of training checkpoints $\mathcal{F}$, and two thresholds $\tau_{\text{conf}}$ and $\tau_{\text{al}}$, we define the category $c(x, y, \mathcal{F})$ as *Discarded* if $\bar{P}_{\mathcal{F}}(x, y) < \tau_{\text{conf}}$ and $v_{al,\mathcal{F}}(x, y) < \tau_{\text{al}}$, and *Selected* otherwise.

Hence, a *Discarded* sample is one for which we have a very low confidence in predicting its associated label whereas we also have low inherent data uncertainty. Finally, given a function $f$ associated with the set of checkpoints $\mathcal{F}$, we define the curated set $D_{\text{curated}} = \{(x, y) | (x, y) \in D_{\text{syn}}, c(x, y, \mathcal{F}) = Selected\}$. We also define $D_{\text{discarded}} = D_{\text{syn}} \setminus D_{\text{curated}}$.

To summarize, the objective of the curation step is that training on the curated synthetic data leads to a better classifier $f_{D_{\text{curated}}}$ for the downstream task, compared to training on the uncurated synthetic data, i.e. $M(f_{D_{\text{curated}}}) > M(f_{\mathcal{D}_{\text{syn}}})$, where $M$ is a performance measure (for example accuracy). In Sec. 3, we empirically show how performance on this curated dataset is superior both for LLM generated data as well as other classes of generative models.

> **Dissecting the role of curation.** We now empirically demonstrate the role of curation in correcting the noisy feature-label relationship present in $D_{\text{syn}}$, highlighting two insights: (i) curation discards samples which are atypical in their label with respect to their neighbors in $D_{\text{syn}}$ (ii) discarded samples can be considered "mislabeled", and we quantify their atypicality using a large held-out dataset $D_{\text{oracle}}$.

▶ **Discarded samples conflict on the label with their neighbors in $D_{\text{syn}}$.** We audit every synthetic sample $(x, y)$ generated by GPT-4 (across 7 datasets) and compute the proportion of its $k$ nearest neighbors in $D_{\text{syn}}$ which share the same label $y$. The agreement with the neighbors assesses the typicality of a sample's $y$ given $x$, where naturally lower agreement is linked to mislabeling, which we aim to detect via curation. Taking $k = 10$, we obtain an average agreement of $a_{\text{curated}} = \mathbf{0.74}$ for $D_{\text{curated}}$, compared to $a_{\text{discarded}} = \mathbf{0.58}$ for $D_{\text{discarded}}$. This shows that the samples removed are those which, despite having similar features $x$, do not agree with their surrounding neighbors' labels. This corroborates ideas in (Ashmore et al., 2021) of how proximity violations are useful to guide remedial action to improve models. Not removing these mislabeled samples injects noise into the downstream classifier, thus reducing performance.

▶ **Assessing discarded samples with $D_{\text{oracle}}$.** Ideally, the samples we select should better align with the true feature-label distribution. Since we don't have access to this distribution explicitly, we compute a proxy for $\eta(x) = \arg\max_y p(Y = y | X = x)$, which we call $\hat{\eta}$. It is obtained by training a classifier on a held-out dataset $D_{\text{oracle}}$—the same size as $D_{\text{test}}$ and an order of magnitude larger than $D_{\text{train}}$. For each synthetic method, we then report the accuracy of $\hat{\eta}$ on both the curated $D_{\text{curated}}$ and discarded $D_{\text{discarded}}$ datasets —see Fig. 5.

We highlight two key observations. First, the curated datasets, for all the generative models, exhibit a higher agreement with the proxy $\hat{\eta}$ than the discarded datasets. This aligns with the desideratum of only keeping samples that exhibit the correct feature-label relationships.

This provides a rationale for why curation helps improve discriminative performance, as samples in $D_{\text{curated}}$ are much more likely to have the correct feature-label relationship.

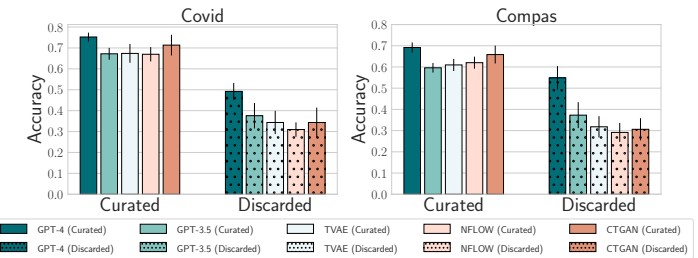

Figure 5: $\hat{\eta}$ aligns more with $D_{\text{curated}}$ than $D_{\text{discarded}}$ for each generative model: the curation step keeps high quality samples tailored to the downstream task.

Second, GPT-4 has a higher agreement with $\hat{\eta}$ on $D_{\text{discarded}}$, compared to the other generators. This illustrates that GPT-4's prior knowledge enables it to better capture the distribution $p(Y|X = x)$. Note that generative baselines (e.g. TVAE) model the joint $p(X, Y)$, *without any context* of which is the set of features and which is the label. In contrast, we can define in the LLM prompt which column is the target $Y$, allowing the LLM to better capture the feature-label relationships. This complements the findings from Fig. 2, which showed that GPT-4 extrapolates to unseen regions of the feature manifold, captured by the support of $p(X)$.

## 3 CURATED LLMs FOR BETTER DATA AUGMENTATION

We now perform an end-to-end quantitative evaluation of CLLM across multiple real-world datasets, for **downstream utility**, demonstrating the value of allying the generative capabilities of LLMs with our curation mechanism. Sec. 3.1 compares the performance of GPT-4 and our curation approach with respect to a variety of state-of-the-art tabular augmentation baselines. Having evaluated CLLM on a range of datasets, we also demonstrate how we can leverage information extracted during curation to characterize datasets via a **hardness proxy**. Sec. 3.2 illustrates how our characterization of samples during the curation step can help to flag synthesized datasets (e.g via the LLM) which, if used for training, will result in poor downstream performance.

**Experimental setup.** We compare CLLM (with GPT-4 (OpenAI, 2023) and GPT-3.5 (Brown et al., 2020)) against a variety of baselines for tabular data generation and augmentation: CTGAN (Xu et al., 2019), TVAE (Xu et al., 2019), Normalizing Flows (Papamakarios et al., 2021), TabDDPM (Kotelnikov et al., 2022), SMOTE (Chawla et al., 2002) and GReaT (Borisov et al., 2023), which fine-tunes an LLM. We evaluate performance on 7 real-world datasets with different feature counts and vary the number of samples available in $D_{\text{train}}$, repeating each experiment across 10 seeds.

While we do not know the exact makeup of the pretraining data of LLMs like GPT-4, there is the possibility that open-source data might be included. This poses a risk of memorization as the primary source of performance gain. To disentangle the role of memorization, we select 4 real-world medical datasets (Maggic (Pocock et al., 2013), Covid (Baqui et al., 2020), SEER (Duggan et al., 2016), CUTRACT (PCUK, 2019)) that require an authorization process to access, hence are unlikely to form part of the LLMs training corpus. We use common open-source datasets (Adult and Drug from the UCI repository (Asuncion & Newman, 2007) and Compas (Angwin et al., 2016)) that are highly reflective of data scarce domains. Further experimental details can be found in Appendix B.

### 3.1 OVERALL PERFORMANCE: DOWNSTREAM UTILITY

We assess overall performance based on *Utility* of the augmented data, which we evaluate in terms of AUC on the real $D_{\text{test}}$, when using four different types of downstream models (see Appendix B). This setup mirrors the widely adopted Train-on-synthetic-Test-on-real (TSTR) (Esteban et al., 2017). Additionally, we compare the performance to training on the small $D_{\text{train}}$, as well as training on the large held-out $D_{\text{oracle}}$, the latter serving as an upper bound.

**GPT-4 + Curation has best overall performance.** Table 3 shows the performance of the proposed CLLM (GPT-4 and GPT-3.5) against baselines — both with and without our curation mechanism. We find that the GPT-4 + Curation variant of CLLM outperforms baselines in almost all settings (20/28). Interestingly, its performance is close to or even exceeds the performance of $D_{\text{oracle}}$. Table 4 further shows that GPT-4 + Curation ranks first on average among all the generative methods.

Table 3: AUC averaged over 4 downstream models on $D_{\text{test}}$ where curation improves performance for all methods across all sample sizes $n$, as indicated by ↑. CLLM w/ GPT-4 (Curated) dataset provides the strongest performance for both private/proprietary datasets and public datasets

| | Real data | | CLLM (OURS) | | | | Baselines | | | | | | | | | | | |
| | | | GPT-4 | | GPT-3.5 | | CTGAN | | TabDDPM | | GReaT | | NFLOW | | SMOTE | | TVAE | |
| Dataset | $D_{\text{oracle}}$ | $D_{\text{train}}$ | Uncur. | Cur. | Uncur. | Cur. | Uncur. | Cur. | Uncur. | Cur. | Uncur. | Cur. | Uncur. | Cur. | Uncur. | Cur. | Uncur. | Cur. |
|---|---|---|---|---|---|---|---|---|---|---|---|---|---|---|---|---|---|---|
| covid (n=20) | 74.41 | 68.50 | 73.78 | **73.87**↑ | 69.85 | 71.41↑ | 59.00 | 63.67↑ | 66.84 | 66.85↑ | 57.38 | 66.46↑ | 62.87 | 68.56↑ | 66.95 | 66.82 | 61.69 | 66.11↑ |
| cutract (n=20) | 72.23 | 70.12 | 71.15 | **72.50**↑ | 69.97 | 71.54↑ | 64.01 | 67.98↑ | 66.05 | 66.59↑ | 52.38 | 67.02↑ | 64.44 | 70.42↑ | 68.41 | 69.24↑ | 68.94 | 70.22↑ |
| maggic (n=20) | 67.41 | 57.13 | 60.70 | **61.48**↑ | 57.54 | 58.69↑ | 52.75 | 54.51↑ | 54.59 | 55.39↑ | 50.29 | 55.64↑ | 54.72 | 57.38↑ | 55.84 | 56.15↑ | 54.08 | 56.19↑ |
| seer (n=20) | 87.92 | 80.67 | 84.53 | **84.82**↑ | 83.34 | 83.71↑ | 74.34 | 78.73↑ | 80.59 | 80.60↑ | 47.57 | 74.43↑ | 76.06 | 79.98↑ | 79.23 | 80.02↑ | 74.53 | 78.73↑ |
| compas (n=20) | 67.51 | 63.11 | **68.01** | 67.91 | 62.07 | 64.43↑ | 55.67 | 62.56↑ | 57.67 | 60.87↑ | 53.33 | 63.59↑ | 59.49 | 64.62↑ | 61.06 | 61.59↑ | 58.30 | 62.58↑ |
| adult (n=20) | 84.17 | 77.45 | 50.39 | 71.48↑ | 49.23 | 72.37↑ | 72.23 | 76.86↑ | 74.35 | 75.04↑ | 67.00 | 77.25↑ | 67.46 | 76.48↑ | 73.75 | 73.67 | 73.20 | 76.90↑ |
| drug (n=20) | 77.81 | 70.84 | 75.08 | **75.29**↑ | 71.68 | 72.14↑ | 68.31 | 72.65↑ | 68.12 | 69.68↑ | 58.78 | 68.89↑ | 62.13 | 67.75↑ | 70.16 | 70.16 | 66.60 | 69.18↑ |
| covid (n=40) | 75.02 | 70.77 | 73.40 | **73.95**↑ | 70.42 | 71.93↑ | 63.63 | 68.46↑ | 70.50 | 70.44 | 56.50 | 68.68↑ | 66.41 | 70.48↑ | 68.66 | 68.44 | 61.03 | 67.35↑ |
| cutract (n=40) | 72.57 | 69.18 | 69.87 | **71.72**↑ | 68.47 | 69.56↑ | 63.01 | 67.87↑ | 65.63 | 67.27↑ | 54.39 | 68.44↑ | 61.40 | 67.98↑ | 67.86 | 67.95↑ | 59.79 | 66.62↑ |
| maggic (n=40) | 67.50 | 58.26 | 59.29 | **60.77**↑ | 57.50 | 59.15↑ | 55.00 | 56.78↑ | 55.24 | 56.94↑ | 48.81 | 56.64↑ | 54.68 | 58.58↑ | 57.40 | 57.44↑ | 55.04 | 57.33↑ |
| seer (n=40) | 87.90 | 82.93 | 84.29 | **84.93**↑ | 83.46 | 84.44↑ | 80.05 | 83.67↑ | 82.59 | 81.37 | 54.93 | 81.11↑ | 79.88 | 84.36↑ | 80.79 | 82.21↑ | 78.69 | 83.62↑ |
| compas (n=40) | 67.35 | 62.34 | 67.57 | **67.85**↑ | 61.34 | 62.84↑ | 56.29 | 61.02↑ | 58.85 | 60.11↑ | 58.88 | 64.37↑ | 58.61 | 63.54↑ | 60.83 | 60.95↑ | 55.94 | 61.04↑ |
| adult (n=40) | 84.43 | 79.44 | 48.31 | 73.82↑ | 49.21 | 74.27↑ | 71.82 | 79.11↑ | 71.51 | 77.99↑ | 66.77 | 78.81↑ | 71.13 | 79.71↑ | 77.90 | 78.84↑ | 72.58 | **80.02**↑ |
| drug (n=40) | 77.71 | 71.86 | 74.30 | **75.79**↑ | 71.33 | 72.76↑ | 69.46 | 72.74↑ | 71.08 | 73.07↑ | 64.89 | 73.64↑ | 62.51 | 70.97↑ | 69.23 | 69.78↑ | 65.22 | 70.30↑ |
| covid (n=100) | 74.52 | 71.57 | 73.77 | **74.71**↑ | 70.71 | 72.76↑ | 69.05 | 72.13↑ | 71.60 | 73.22↑ | 63.52 | 72.04↑ | 64.25 | 72.64↑ | 70.08 | 70.78↑ | 69.05 | 71.96↑ |
| cutract (n=100) | 72.36 | 70.96 | 70.20 | **72.51**↑ | 69.97 | 71.94↑ | 67.94 | 72.42↑ | 70.53 | 71.98↑ | 55.72 | 69.14↑ | 67.59 | 72.42↑ | 68.79 | 69.68↑ | 66.89 | 71.52↑ |
| maggic (n=100) | 67.46 | 59.65 | 58.98 | **61.32**↑ | 55.71 | 58.90↑ | 57.20 | 59.34↑ | 57.26 | 58.28↑ | 49.54 | 57.91↑ | 56.36 | 60.11↑ | 58.89 | 58.99↑ | 56.17 | 58.86↑ |
| seer (n=100) | 87.79 | 83.95 | 84.45 | **85.37**↑ | 83.92 | 85.08↑ | 81.60 | 85.14↑ | 83.04 | 84.83↑ | 70.32 | 83.83↑ | 81.16 | 85.03↑ | 81.82 | 82.49↑ | 78.88 | 84.50↑ |
| compas (n=100) | 67.18 | 62.56 | 68.02 | **68.19**↑ | 60.10 | 62.47↑ | 60.01 | 63.73↑ | 58.32 | 61.34↑ | 59.97 | 64.19↑ | 60.02 | 64.04↑ | 61.44 | 61.73↑ | 59.97 | 62.82↑ |
| adult (n=100) | 84.34 | 81.24 | 46.09 | 74.57↑ | 47.56 | 73.97↑ | 74.29 | 80.45↑ | 75.93 | 78.22↑ | 77.09 | **81.66**↑ | 70.70 | 81.04↑ | 80.56 | 81.10↑ | 74.04 | 80.23↑ |
| drug (n=100) | 78.00 | 73.58 | 76.24 | **76.74**↑ | 69.46 | 71.05↑ | 68.19 | 73.28↑ | 72.43 | 73.79↑ | 67.26 | 75.28↑ | 62.67 | 73.12↑ | 70.90 | 71.53↑ | 68.22 | 73.59↑ |
| covid (n=200) | 74.69 | 72.33 | 73.40 | **74.62**↑ | 70.70 | 73.12↑ | 71.07 | 73.89↑ | 72.47 | 74.44↑ | 65.55 | 73.07↑ | 65.04 | 72.90↑ | 71.68 | 71.87↑ | 67.89 | 72.38↑ |
| cutract (n=200) | 72.52 | 71.75 | 71.39 | 73.01↑ | 70.28 | 72.39↑ | 69.28 | 72.41↑ | 71.83 | **74.03**↑ | 66.66 | 72.49↑ | 68.77 | 73.16↑ | 70.23 | 70.80↑ | 66.61 | 71.87↑ |
| maggic (n=200) | 67.37 | 61.39 | 58.92 | **61.41**↑ | 57.33 | 60.16↑ | 58.48 | 61.33↑ | 56.26 | 57.20↑ | 50.74 | 59.60↑ | 55.95 | 60.75↑ | 60.73 | 60.78↑ | 57.18 | 60.23↑ |
| seer (n=200) | 87.84 | 84.63 | 84.39 | 85.56↑ | 83.48 | 84.80↑ | 82.04 | 85.34↑ | 84.39 | **86.57**↑ | 82.15 | 86.03↑ | 77.73 | 85.19↑ | 83.38 | 84.15↑ | 79.71 | 85.26↑ |
| compas (n=200) | 67.14 | 63.27 | 67.02 | **68.15**↑ | 60.48 | 63.39↑ | 60.58 | 64.32↑ | 60.60 | 63.52↑ | 61.11 | 65.08↑ | 56.58 | 63.60↑ | 61.99 | 62.80↑ | 60.15 | 63.99↑ |
| adult (n=200) | 84.25 | 82.12 | 40.96 | 75.84↑ | 49.89 | 72.81↑ | 78.18 | 82.32↑ | 81.66 | 83.17↑ | 80.06 | **83.32**↑ | 74.31 | 82.64↑ | 82.26 | 82.39↑ | 75.21 | 82.02↑ |
| drug (n=200) | 77.36 | 76.10 | 75.58 | 76.06↑ | 70.66 | 72.81↑ | 71.31 | 75.98↑ | 69.61 | 71.79↑ | 72.35 | **77.41**↑ | 65.25 | 75.26↑ | 74.38 | 74.78↑ | 68.39 | 74.33↑ |

**Sample size sensitivity.** We now investigate the performance gains of CLLM as we vary the number of samples $n$ in $D_{\text{train}}$, in Table 3 and Table 4. Performance improvements and high ranking across datasets for CLLM (GPT-4+Curation) are especially noticeable in the ultra low-data regime (i.e. $n < 100$). In this regime, the limited size of $D_{\text{train}}$ severely constrains the other baseline methods. In contrast, as illustrated in Sec. 2.1, CLLM can leverage GPT-4's prior knowledge to extrapolate beyond the small $D_{\text{train}}$, thereby improving downstream performance. As expected, the performance gap between CLLM and other methods decreases as the size of $D_{\text{train}}$ grows (e.g. $n = 200$), where sufficient training data helps other generators achieve good performance.

**Curation generally helps all generative models.** Our curation mechanism consistently benefits all generative models for the different $n$. It ensures only high quality samples are retained, which is crucial for good data augmentation and downstream performance and has been overlooked in previous works. This explains why the combination of the best generative model and curation, which is CLLM, gives the best results and highest rankings in the low-data regime (e.g. $n = 20$).

Table 4: Average rank of approaches across the different datasets and seeds. CLLM w/ GPT-4 ranks first across all $n$ and curation improves all the generative models.

| Method | n=20 | n=40 | n=100 | n=200 |
|---|---|---|---|---|
| CLLM w/ GPT-4 | **2.71 ± 1.44** | **2.14 ± 1.06** | **2.29 ± 1.19** | **3.29 ± 1.38** |
| GPT-4 | 3.86 ± 1.73 | 4.29 ± 1.83 | 6.00 ± 1.77 | 7.57 ± 1.65 |
| CLLM w/ GPT-3.5 | 4.14 ± 0.94 | 4.14 ± 0.71 | 6.86 ± 1.24 | 7.57 ± 0.70 |
| NFLOW (curated) | 6.00 ± 1.21 | 4.71 ± 0.80 | 4.00 ± 0.57 | 4.71 ± 0.63 |
| GPT-3.5 | 6.71 ± 1.52 | 7.29 ± 1.26 | 11.57 ± 0.94 | 12.57 ± 0.57 |
| TVAE (curated) | 7.14 ± 1.17 | 7.86 ± 1.30 | 6.43 ± 0.40 | 6.71 ± 0.52 |
| SMOTE (curated) | 7.71 ± 0.33 | 8.14 ± 0.91 | 7.71 ± 1.19 | 7.43 ± 1.07 |
| SMOTE | 7.86 ± 0.55 | 9.57 ± 0.80 | 9.57 ± 1.09 | 9.00 ± 1.03 |
| TabDDPM (curated) | 8.29 ± 0.98 | 8.00 ± 0.93 | 6.00 ± 0.95 | 5.14 ± 1.68 |
| CTGAN (curated) | 8.29 ± 1.42 | 7.14 ± 0.91 | 4.14 ± 0.62 | 3.71 ± 0.39 |
| GReaT (curated) | 8.57 ± 1.50 | 6.57 ± 1.21 | 6.29 ± 1.38 | 3.57 ± 0.92 |
| TabDDPM | 10.14 ± 1.19 | 9.86 ± 1.15 | 10.00 ± 1.03 | 10.29 ± 1.02 |
| TVAE | 12.14 ± 0.89 | 14.00 ± 0.70 | 13.71 ± 0.39 | 14.43 ± 0.40 |
| NFLOW | 12.86 ± 0.47 | 14.14 ± 0.37 | 14.00 ± 0.45 | 15.29 ± 0.33 |
| CTGAN | 13.86 ± 0.68 | 13.14 ± 0.47 | 12.86 ± 0.37 | 12.00 ± 0.53 |
| GReaT | 15.71 ± 0.26 | 15.00 ± 0.53 | 14.57 ± 1.03 | 12.71 ± 0.96 |

**Performance benefits maintained for private and public datasets.** One may hypothesize that the strong LLM (e.g. GPT-4) performance is explained by datasets being part of the LLMs' training corpus, hence possibly being memorized. We show in Table 3 that it is unlikely, as we retain strong performance for both open-source datasets, as well as private medical datasets which require authorization processes for access and are unlikely to be part of the LLM pretraining dataset.

*Remark on ICL versus fine-tuning.* Our results in Table 3 and Table 4 indicate that ICL is better than fine-tuning (GReaT baseline) in the low-data regime. This highlights the difficulty of fine-tuning in this regime, where it is easy to overfit to $D_{\text{train}}$. As we increase the number of samples, this baseline coupled with curation improves toward the level of CLLM (GPT-4).

### 3.2 HARDNESS: A PROXY SIGNAL TO FLAG POOR QUALITY SYNTHETIC DATASETS

Having a systematic way to assess datasets generated by LLMs like GPT-4 is important because their black-box nature provides little control on their generation quality. This contrasts conventional generators for which training loss is an exploitable signal. Hence, we ask: could we have a signal to identify a potential problematic dataset generated by GPT-4 without an exhaustive manual review? For example, GPT-4 produced low-quality synthetic data for the Adult dataset (across the different sample sizes) resulting in poor downstream performance. While curation improves it, downstream performance is still suboptimal. Addressing this question is important, since datasets are rarely created by the ML model builder in real-world ML workflows, but rather by specialist data teams or data owners (Gebru et al., 2021; Sambasivan et al., 2021; Goncalves et al., 2020). Hence, having a signal to preemptively flag a potentially suboptimal generated dataset spares investment in both storing the subpar data and/or training a model likely to underperform on real data.

$D_{\text{syn}}$ should intuitively be considered imperfect if curation discards many of its samples, since the number of discarded samples measures the quality of samples with respect to the small but gold-standard $D_{\text{train}}$. Hence, we investigate the relationship between test performance (AUC) and the proportion of samples discarded by the curation. Fig. 6, where each point is a synthetic dataset generated by GPT-4 (e.g. Adult, Compas), shows a strong negative linear relationship between these two quantities. This holds across the different $n$ with slopes fairly stable around $-1.4$. This relationship corroborates the poor quality of the dataset generated by GPT-4 on the Adult dataset, providing a useful proxy that $D_{\text{syn}}$ is unlikely to lead to good downstream performance.

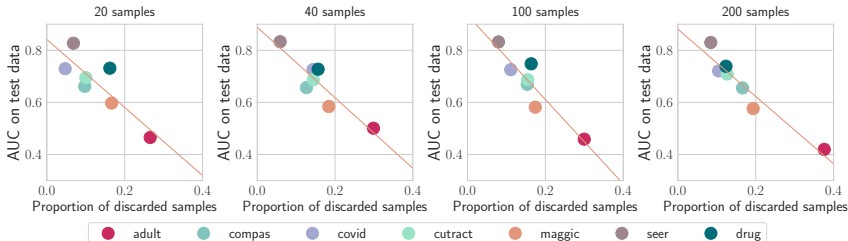

Figure 6: The proportion of discarded samples $D_{\text{syn}}$ is a proxy for test performance. This negative linear relationship where each point is a synthetic dataset generated by GPT-4 (e.g. Adult, Covid, Compas) allows us to flag datasets that will lead to unreliable downstream performance.

## 4 DISCUSSION

We introduce CLLM, an approach for data augmentation in the ultra low-data setting. CLLM exploits the prior knowledge of LLMs along with curation for improved downstream performance. As empirically shown, CLLM outperforms traditional generative models—most noticeably on underrepresented subgroups, for which data augmentation is of utmost importance. CLLM is grounded in the ICL capability of LLMs, and benefits from its simplicity. We studied GPT-3.5 and GPT-4 as backbones for CLLM. The cost of the API access pose limitations, e.g. on wide accessibility, on knowing which data was used for training the models, and on understanding the LLM's output better. Using smaller and open LLMs could overcome these limitations, though this could come with a reduction in performance. We leave this as a promising direction for future work. Further improvements may be achieved through different tuning and prompting of the LLM, as shown in different domains (Meng et al., 2023; Liu et al., 2023). Improving LLM tuning and prompting is beyond the scope of our work, but we regard this as a promising avenue for future work.

Data scarcity and computational limitations are deterrents for developing ML. These challenges should inspire cutting-edge ML research (De-Arteaga et al., 2018). We believe CLLM takes a step in this direction toward improving the use of ML in low-data settings, across **society** (e.g. underrepresented subgroups (Suresh & Guttag, 2021)), **domains** (e.g. healthcare (Alami et al., 2020; Owoyemi et al., 2020)) and **regions** (e.g. LMICs).

## ETHICS AND REPRODUCIBILITY STATEMENTS

**Ethics.** In this work, we evaluate CLLM using multiple real-world datasets. The private datasets are *de-identified* and used in accordance with the guidance of the respective data providers. We follow recommendations to use the Azure OpenAI service when using GPT-4 and GPT-3.5 models, where via the agreement we ensure the medical data is not sent for human review or stored, hence respecting the guidelines given by the dataset providers. LLMs may make errors and may reflect or exacerbate societal biases that are present in their data (Li et al., 2023). Though the curation in CLLM improves synthetic data quality, it does not directly aim to remove biases. The quality and fairness of generated data should always be evaluated. More research into LLM bias is required before methods like CLLM should be applied to real-world sensitive settings like healthcare and finance. Finally, increasing access to ML across regions, domains and societies is also about more than just technology. We believe broader engagement and discussion with various stakeholders is crucial to responsibly expand ML access, thereby realizing the benefits of ML in an equitable way.

**Reproducibility.** Experiments are described in Section 4 with further details of the method, experimental setup and datasets included in Appendix B. Code will be released upon acceptance.

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
