# Appendix: Curated LLM: Synergy of LLMs and Data Curation for tabular augmentation in ultra low-data regimes

## Table of Contents

# A    EXTENDED RELATED WORK

This paper primarily engages with the work on data augmentation when we have limited data, where our primary goal is synthetic data generation to augment the dataset. Generating synthetic datasets not only helps improve downstream performance, but it is also a flexible solution as it doesn't tie the data consumer to any particular downstream model. That said, beyond the major difference of synthetic data generation, for completeness we contrast our setting of learning from limited data with other seemingly similar settings and highlight their differences.

**Contrasting learning w/ limited data vs other settings.** The challenge of learning from limited data, while seemingly related to several other learning paradigms, presents distinct differences and unique intricacies that warrant dedicated study.

*Transfer learning* (Pan & Yang, 2009), *domain adaptation* (Farahani et al., 2021), and *few-shot learning* (Wang et al., 2020) employ additional data resources or rely on specific task-related assumptions to improve learning performance. These methods exploit large labeled data from a source domain, unlabeled data in a target domain, or leverage knowledge from related tasks respectively. For example, Levin et al. (2022) and Jin & Ucar (2023) use models trained on labeled data from a source domain, while Ruiz et al. (2023) and Margeloiu et al. (2022) leverage knowledge-graphs. This is in contrast to our setting, considered of learning with limited data, which must function with whatever scarce labeled data it has, without making any assumptions about the availability of additional data or tasks.

*Active learning* (Settles, 2009) and *semi-supervised learning* (van Engelen & Hoos, 2019; Chapelle et al., 2006) also operate under the premise of having access to plentiful unlabeled data and the capacity to interactively query labels. However, in our setting, considered learning with limited data does not inherently assume such capabilities, focusing instead on limited labeled data only.

Furthermore, active learning primarily focuses on the iterative process of selecting data samples that, when labeled, are expected to most significantly improve the model's performance. This selection is typically based on criteria such as uncertainty sampling which focuses on **epistemic uncertainty** (Mussmann & Liang, 2018; Houlsby et al., 2011; Kirsch et al., 2019; Nguyen et al., 2022). The primary objective is to minimize labeling effort while maximizing the model's learning efficiency. Additionally, active learning would aims to label instances based on epistemic uncertainty where the model struggles to make accurate predictions, yet the samples themselves are correct. In contrast, CLLM leverage training dynamics based on **aleatoric uncertainty** and confidence and is designed to discard samples that might jeopardize the downstream accuracy. These samples can be considered to have inherent issues or are erroneous, such as being "mislabeled". To summarize, in active learning, epistemic uncertainty is used to identify data points that, if labeled, would yield the most significant insights for model training. In our approach, they serve to identify and exclude/filter data points that could potentially deteriorate the model's performance.

*Self-supervised learning* (Liu et al., 2021) leverages large amounts of unlabeled data to learn useful representations for downstream tasks. However, in our setting, considered learning with limited data does not inherently assume such access to vast amounts of unlabeled data.

**Data-centric AI.** Ensuring high data quality is a critical but often overlooked problem in ML, where the focus is optimizing models (Sambasivan et al., 2021). Even when it is considered, the process of assessing datasets is adhoc or artisanal (Seedat et al., 2022b). However, the recent push of data-centric AI (Liang et al., 2022; Polyzotis & Zaharia, 2021; Zha et al., 2023) aims to develop systematic tools to curate existing datasets. Our work contributes to this nascent body of work (Seedat et al., 2023) – presenting CLLM, which, to the best of our knowledge, is the first systematic data-centric framework looking at how we can tailor synthetic datasets) rather than real datasets) to downstream task use with data curation.

**Why Data Augmentation?** Data augmentation is a flexible approach to address the ultra low-data regime. An alternative might be to resort to a pretrained black-box model for classification, which could be for example via in-context learning for classification (Dong et al., 2022). However, such a solution is inadequate for several reasons, many of which would prevent real-world utility (e.g. in LMICs):

▶ *Not economical over the long term:* While using an LLM like GPT for classification may seem attractive due to its few-shot capabilities, it is likely not economically viable in real-world settings, especially in LMICs. The reason is classifying each sample will incur a cost to call the LLM, hence scales linearly with the number of test samples. Over time, the cumulative cost of these calls will surpass the once-off fixed cost associated with generating data. With data augmentation, once the dataset is augmented, there are no additional deployment time costs associated with the LLM. Indeed, the downstream models e.g. a random forest or XGBoost have negligible inference costs.

▶ *Control, interpretability and auditability:* Relying on a large, pre-trained LLM as a black-box classifier raises several concerns. (1) we have no control over our downstream classifier and its architecture, (2) lack of interpretability and auditability of the LLM when issuing predictions. In contrast, training a downstream model on augmented data maintains the ability to understand and explain how the model is making decisions (e.g. feature importance). This is especially crucial in contexts where accountability, transparency, and validation of machine learning processes are paramount.

▶ *Independence and self-sufficiency:* Relying on third-party services for continuous classification means being dependent on their availability, pricing models, and potential changes in the LLM version. By augmenting data and training a downstream classifier on the augmented dataset, we ensure that there is no external dependencies such as increasing costs or reduced performance with LLM version updates.

▶ *Hardware and financial constraints:* Even if we opt for an open-source LLM (e.g. Falcon (Penedo et al., 2023) or LLaMA-2 (Touvron et al., 2023)), deploying and running it locally demands significant computational resources. Typically, these models require GPUs with high amounts of VRAM for optimal performance (e.g. needing around 40 GB hencing requiring an A100 GPU for Falcon-40b and LLaMA-2 65B). Such high-end GPUs are expensive, and are likely to be inaccessible in a LMIC setting. Furthermore, renting hardware by the hour can quickly become prohibitively expensive. Data augmentation, on the other hand, can often be performed on modest hardware, and once the augmented dataset is created, many classifiers can be trained without the need for high-end GPUs, making the entire process more financially accessible.

In conclusion, while large language models offer vast knowledge, for low-data settings in low-income countries, data augmentation provides a more cost-effective, controllable, and interpretable solution for building robust classifiers.

# B  EXPERIMENTAL DETAILS

We provide details on our datasets used, as well as, other experimental specifics including: generation, curation, downstream model, prompt template.

## B.1  DATASETS

We summarize the different datasets we use in this paper in Table 5. The datasets vary in number of samples, number of features and domain.

Table 5: Summary of the datasets used. * Denotes private/proprietary datasets.

| Name | $n$ samples | $n$ features | Domain |
|---|---|---|---|
| Adult Income (Asuncion & Newman, 2007) | 30k | 12 | Finance |
| Compas (Angwin et al., 2016) | 5k | 13 | Criminal justice |
| *Covid-19 (Baqui et al., 2020) | 7k | 29 | Healthcare/Medicine |
| *CUTRACT Prostate (PCUK, 2019) | 2k | 12 | Healthcare/Medicine |
| Drug (Fehrman et al., 2017) | 2k | 27 | Healthcare/Medicine |
| *MAGGIC (Pocock et al., 2013) | 41k | 29 | Healthcare/Medicine |
| *SEER Prostate (Duggan et al., 2016) | 20k | 12 | Healthcare/Medicine |

We detail the dataset splits used in Sec. 3.1. For each dataset and number of samples $n \in \{20, 40, 100, 200\}$, we sample a training set $D_{\text{train}}$ such that $|D_{\text{train}}| = n$, and each target class has the same number of samples. We then split the remaining samples into two non-overlapping datasets, $D_{\text{oracle}}$ and $D_{\text{test}}$, which have the same cardinality. This procedure is repeated $n_{\text{seed}} = 10$ times, thus leading to different training and test sets. Note that the different generative models use the same $D_{\text{train}}$ and $D_{\text{test}}$ for a given seed.

**Motivation for the choice of datasets.**

1. **Open-source:** Adult, Drug and Compas are widely used open-source datasets used in the tabular data literature. Adult and Drug are both UCI datasets that have been used in many papers, while Compas is part of OpenML Vanschoren et al. (2013). Our reason for selecting them is that, despite them being open-source, they are highly reflective of domains in which we might be unable to collect many samples — hence in reality would often be in an ultra-low data regime.

2. **Private datasets:** We wanted to disentangle the possible role of memorization in the strong performance of the LLM. To ensure the datasets are not in the LLMs training corpus, we selected 4 private medical datasets that need an authorization process to access. Hence, these datasets would not be part of the LLMs training corpus given their proprietary nature and hence would be unseen to the LLM. While the private and unseen aspect was the main motivation, we also wish to highlight that these are real-world medical datasets. Consequently, this allows us to test a highly realistic problem setting.

## B.2  DATA GENERATION.

**GPT-4 and GPT-3.5** We access GPT-4 (OpenAI, 2023) and GPT-3.5-Turbo (Brown et al., 2020) through the API. We use a temperature of 0.9.

**GReaT.** GReaT Borisov et al. (2023) is a generative model which fine-tunes an LLM based on a training set. We use the implementation provided by authors.

**Generative model based approaches.** For the other baselines used in 3.1, we use the library SynthCity (Qian et al., 2023), using the defaults. We detail each next.

- TVAE: this is a conditional Variational Auto Encoder (VAE) for tabular data and is based on Xu et al. (2019)

- CTGAN: A conditional generative adversarial network which can handle tabular data and is based on Xu et al. (2019)

- NFLOW: Normalizing Flows are generative models which produce tractable distributions where both sampling and density evaluation can be efficient and exact.

- TabDDPM: A diffusion model that can be universally applied to any tabular dataset, handles any type of feature and is based on Kotelnikov et al. (2022)

**Traditional Data Augmentation.** We use SMOTE (Chawla et al., 2002) which augments data by considering nearest neighbors and performing linear interpolations. We use the implementation provided by Lemaître et al. (2017), and set the number of neighbors $k$ to 5.

### B.3 DATA CURATION

**Learning dynamics computation** We train an XGBoost with 100 estimators on $D_{\text{train}}$. We then compute predictive confidence and aleatoric uncertainty for the samples in $D_{\text{syn}}$. The motivation for the choice of an XGBoost backbone is that we cannot expect good performance by choosing "any" curation model, but rather we require a curation model with enough capacity and generalization properties — where boosting methods like XGBoost used in our work have shown to achieve best performance on tabular data. This leads to our guideline for the curation step: the model used for curation should be **at least as flexible** as the model that the practitioner intends to use for the downstream task.

**Learning dynamics thresholds** Recall that CLLM has two thresholds $\tau_{\text{conf}}$ and $\tau_{\text{al}}$ on the predictive confidence and aleatoric uncertainty respectively, as defined in 2.2. We set $\tau_{\text{conf}} = 0.2$, in order to select high confidence samples. We adopt an adaptive threshold for $\tau_{\text{al}}$ based on the dataset, such that $\tau_{\text{al}} = 0.75 \cdot (\max(v_{al}(D_{\text{syn}})) - \min(v_{al}(D_{\text{syn}})))$. Note that by definition $v_{al}(D_{\text{syn}})$ is bounded between 0 and 0.25.

**Example of learning dynamics** We include examples of learning dynamics computed for 20 samples in Fig. 7.

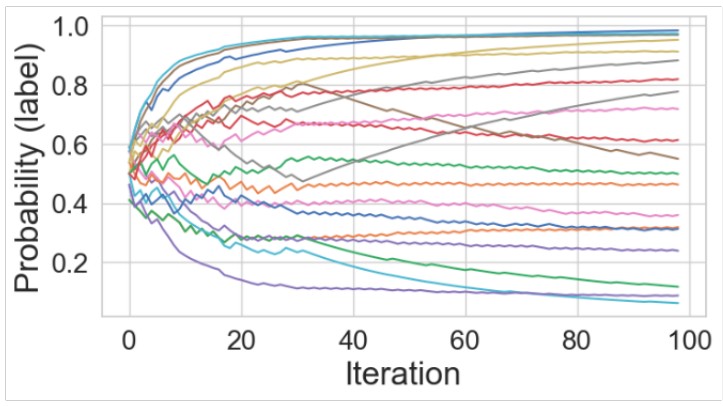

Figure 7: Learning dynamics computed for 20 samples

### B.4 DOWNSTREAM TASK

We compute downstream performance in Sec. 3.1 using four different downstream models: XG-Boost, Random Forest, Decision tree, and Logistic Regression.

## B.5 PROMPT EXAMPLE

We include the template of the prompts used throughout the paper. We show how we include (1) in-context examples (demonstrations), (2) contextual information including dataset background and feature information and (3) the instruction.

```
1    System role: 'You are a tabular synthetic data generation model.'
2
3    You are a synthetic data generator.
4    Your goal is to produce data which mirrors \
5    the given examples in causal structure and feature and label
     distributions \
6    but also produce as diverse samples as possible.
7
8    I will give you real examples first.
9
10   Context: Leverage your medical knowledge about covid and Brazil to
     generate 1000 realistic but diverse samples.
11
12   example data: {data}
13
14   The output should be a markdown code snippet formatted in the
     following schema:
15
16   "Sex_male": string  // feature column
17   "Age": string  // feature column
18   "Age_40": string  // feature column
19   "Age_40_50": string  // feature column
20   "Age_50_60": string  // feature column
21   "Age_60_70": string  // feature column
22   "Age_70": string  // feature column
23   "Fever": string  // feature column
24   "Cough": string  // feature column
25   "Sore_throat": string  // feature column
26   "Shortness_of_breath": string  // feature column
27   "Respiratory_discomfort": string  // feature column
28   "SPO2": string  // feature column
29   "Dihareea": string  // feature column
30   "Vomitting": string  // feature column
31   "Cardiovascular": string  // feature column
32   "Asthma": string  // feature column
33   "Diabetis": string  // feature column
34   "Pulmonary": string  // feature column
35   "Immunosuppresion": string  // feature column
36   "Obesity": string  // feature column
37   "Liver": string  // feature column
38   "Neurologic": string  // feature column
39   "Renal": string  // feature column
40   "Branca": string  // feature column
41   "Preta": string  // feature column
42   "Amarela": string  // feature column
43   "Parda": string  // feature column
44   "Indigena": string  // feature column
45   "is_dead": string  // label if patient dead or not, is_dead
46
47   DO NOT COPY THE EXAMPLES but generate realistic but new and diverse
     samples which have the correct label conditioned on the features.
```

Listing 1: Template of the prompt

# C  ADDITIONAL RESULTS

## C.1  DETAILED RESULTS FOR SECTION 3.1

We report additional results for Sec. 3.1, showing the AUC for each downstream model used (XG-Boost, Random forest, Logistic regression, Decision tree). As we can see, the conclusion that curation helps improve downstream performance holds for each of these various downstream models, as is indicated by the green arrows in Tables 6, 7, 8, 9.

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

## C.2 FULL RESULTS FOR PERFORMANCE EVALUATION

We report full results with standard deviation for the results from the main paper. The performance is AUC averaged over XGBoost, Random forest, Logistic regression, Decision tree.

Table 10: AUC averaged over 4 downstream models on $D_{\text{test}}$ where curation improves performance for all methods across all sample sizes $n$, as indicated by ↑. CLLM w/ GPT-4 (Curated) dataset provides the strongest performance for both private/proprietary datasets and public datasets

| | Real data | | CLLM (OURS) | | | | Baselines | | | | | | | | | | | |
| | $D_{\text{oracle}}$ | $D_{\text{train}}$ | GPT-4 | | GPT-3.5 | | CTGAN | | TabDDPM | | GReaT | | NFLOW | | SMOTE | | TVAE | |
| Dataset | | | Uncur. | Cur. | Uncur. | Cur. | Uncur. | Cur. | Uncur. | Cur. | Uncur. | Cur. | Uncur. | Cur. | Uncur. | Cur. | Uncur. | Cur. |
|---|---|---|---|---|---|---|---|---|---|---|---|---|---|---|---|---|---|---|
| covid (n=20) | $74.41_{(0.11)}$ | $68.50_{(1.57)}$ | $73.78_{(0.31)}$ | $\mathbf{73.87}_{(0.50)}$ | $69.85_{(0.75)}$ | $71.41_{(0.92)}$ | $59.00_{(2.25)}$ | $63.67_{(2.51)}$ | $66.84_{(1.66)}$ | $66.85_{(1.56)}$ | $57.38_{(1.47)}$ | $66.46_{(0.80)}$ | $62.87_{(0.98)}$ | $68.56_{(1.07)}$ | $66.95_{(1.66)}$ | $66.82_{(1.89)}$ | $61.69_{(2.72)}$ | $66.11_{(2.79)}$ |
| cutract (n=20) | $72.23_{(0.65)}$ | $70.12_{(1.16)}$ | $71.15_{(0.46)}$ | $\mathbf{72.50}_{(0.76)}$ | $69.97_{(1.13)}$ | $71.54_{(1.38)}$ | $64.01_{(2.26)}$ | $67.98_{(1.63)}$ | $66.05_{(1.14)}$ | $66.59_{(1.49)}$ | $52.38_{(1.78)}$ | $67.02_{(0.97)}$ | $64.44_{(1.05)}$ | $70.42_{(1.25)}$ | $68.41_{(1.34)}$ | $69.24_{(1.25)}$ | $68.94_{(1.38)}$ | $70.22_{(1.12)}$ |
| maggic (n=20) | $67.41_{(0.06)}$ | $57.13_{(0.85)}$ | $60.70_{(0.38)}$ | $\mathbf{61.48}_{(0.49)}$ | $57.54_{(0.33)}$ | $58.69_{(0.76)}$ | $52.75_{(1.34)}$ | $54.51_{(1.40)}$ | $54.59_{(1.02)}$ | $55.39_{(1.07)}$ | $50.29_{(0.52)}$ | $55.64_{(0.43)}$ | $54.72_{(1.38)}$ | $57.38_{(1.06)}$ | $55.84_{(1.21)}$ | $56.15_{(1.20)}$ | $54.08_{(1.05)}$ | $56.19_{(0.76)}$ |
| seer (n=20) | $87.92_{(0.08)}$ | $80.67_{(1.67)}$ | $84.53_{(0.33)}$ | $\mathbf{84.82}_{(0.47)}$ | $83.34_{(0.95)}$ | $83.71_{(0.52)}$ | $74.34_{(1.11)}$ | $78.73_{(1.13)}$ | $80.59_{(1.32)}$ | $80.60_{(1.32)}$ | $47.57_{(2.51)}$ | $74.43_{(3.36)}$ | $76.06_{(1.68)}$ | $79.98_{(1.65)}$ | $79.23_{(2.17)}$ | $80.02_{(2.08)}$ | $74.53_{(1.37)}$ | $78.73_{(1.24)}$ |
| compas (n=20) | $67.51_{(0.03)}$ | $63.11_{(0.96)}$ | $\mathbf{68.01}_{(0.04)}$ | $67.91_{(0.55)}$ | $62.07_{(1.51)}$ | $64.43_{(1.38)}$ | $55.67_{(1.60)}$ | $62.56_{(1.61)}$ | $57.67_{(1.90)}$ | $60.87_{(1.33)}$ | $53.33_{(1.85)}$ | $63.59_{(1.30)}$ | $59.49_{(1.72)}$ | $64.62_{(1.14)}$ | $61.06_{(1.92)}$ | $61.59_{(2.15)}$ | $58.30_{(1.77)}$ | $62.58_{(1.87)}$ |
| adult (n=20) | $84.17_{(0.10)}$ | $77.45_{(1.25)}$ | $50.39_{(3.99)}$ | $71.48_{(2.34)}$ | $49.23_{(2.57)}$ | $72.37_{(2.26)}$ | $72.23_{(1.26)}$ | $76.86_{(1.25)}$ | $74.35_{(1.48)}$ | $75.04_{(1.82)}$ | $67.00_{(4.72)}$ | $77.25_{(1.49)}$ | $67.46_{(3.76)}$ | $76.45_{(1.77)}$ | $73.75_{(1.64)}$ | $73.67_{(1.46)}$ | $73.20_{(1.51)}$ | $76.90_{(1.64)}$ |
| drug (n=20) | $77.81_{(0.55)}$ | $70.84_{(2.25)}$ | $75.08_{(1.17)}$ | $\mathbf{75.29}_{(1.11)}$ | $71.68_{(2.25)}$ | $72.14_{(2.64)}$ | $68.31_{(2.81)}$ | $72.65_{(2.00)}$ | $68.12_{(2.38)}$ | $69.68_{(2.41)}$ | $58.78_{(4.26)}$ | $68.89_{(3.57)}$ | $62.13_{(4.94)}$ | $67.75_{(3.72)}$ | $70.16_{(1.87)}$ | $70.16_{(1.75)}$ | $66.60_{(2.95)}$ | $69.18_{(2.59)}$ |
| covid (n=40) | $75.02_{(0.22)}$ | $70.77_{(0.96)}$ | $73.40_{(0.61)}$ | $\mathbf{73.95}_{(0.67)}$ | $70.42_{(0.92)}$ | $71.93_{(0.60)}$ | $63.63_{(1.15)}$ | $68.46_{(0.93)}$ | $70.50_{(1.49)}$ | $70.44_{(1.37)}$ | $56.50_{(0.83)}$ | $68.68_{(1.39)}$ | $66.41_{(2.51)}$ | $70.48_{(1.48)}$ | $68.66_{(1.48)}$ | $68.44_{(1.28)}$ | $61.03_{(0.96)}$ | $67.35_{(0.60)}$ |
| cutract (n=40) | $72.57_{(0.48)}$ | $69.18_{(0.65)}$ | $69.87_{(0.62)}$ | $\mathbf{71.72}_{(0.46)}$ | $68.47_{(0.59)}$ | $69.56_{(0.50)}$ | $63.01_{(2.33)}$ | $67.87_{(1.36)}$ | $65.63_{(2.71)}$ | $67.27_{(2.38)}$ | $54.39_{(1.59)}$ | $68.44_{(0.41)}$ | $61.40_{(1.95)}$ | $67.98_{(1.06)}$ | $67.86_{(0.72)}$ | $67.95_{(0.59)}$ | $59.79_{(1.85)}$ | $66.62_{(1.06)}$ |
| maggic (n=40) | $67.50_{(0.04)}$ | $58.26_{(0.55)}$ | $59.29_{(0.50)}$ | $\mathbf{60.77}_{(0.36)}$ | $57.50_{(0.69)}$ | $59.15_{(0.48)}$ | $55.00_{(1.27)}$ | $56.78_{(1.17)}$ | $55.24_{(0.63)}$ | $56.94_{(0.50)}$ | $48.81_{(0.73)}$ | $56.64_{(0.63)}$ | $54.68_{(0.53)}$ | $58.58_{(0.54)}$ | $57.40_{(0.57)}$ | $57.44_{(0.67)}$ | $55.04_{(1.14)}$ | $57.33_{(1.06)}$ |
| seer (n=40) | $87.90_{(0.09)}$ | $82.93_{(0.55)}$ | $84.29_{(0.39)}$ | $\mathbf{84.93}_{(0.46)}$ | $83.46_{(1.20)}$ | $84.44_{(0.69)}$ | $80.05_{(0.93)}$ | $83.67_{(0.51)}$ | $82.59_{(1.48)}$ | $81.37_{(1.11)}$ | $54.93_{(2.06)}$ | $81.11_{(1.24)}$ | $79.88_{(0.53)}$ | $84.36_{(0.62)}$ | $80.79_{(0.67)}$ | $82.21_{(0.67)}$ | $78.69_{(2.32)}$ | $83.62_{(1.06)}$ |
| compas (n=40) | $67.35_{(0.16)}$ | $62.34_{(0.79)}$ | $67.57_{(0.40)}$ | $\mathbf{67.85}_{(0.37)}$ | $61.34_{(1.67)}$ | $62.84_{(1.21)}$ | $56.29_{(1.96)}$ | $61.02_{(1.69)}$ | $58.85_{(1.28)}$ | $60.11_{(1.22)}$ | $58.88_{(1.04)}$ | $64.37_{(0.91)}$ | $58.61_{(1.35)}$ | $63.54_{(1.11)}$ | $60.83_{(1.19)}$ | $60.95_{(1.22)}$ | $55.94_{(1.50)}$ | $61.04_{(1.56)}$ |
| adult (n=40) | $84.43_{(0.06)}$ | $79.44_{(1.03)}$ | $48.31_{(3.58)}$ | $73.82_{(2.19)}$ | $49.21_{(1.78)}$ | $74.27_{(1.49)}$ | $71.82_{(1.22)}$ | $79.11_{(0.71)}$ | $71.51_{(1.85)}$ | $77.99_{(0.65)}$ | $66.77_{(3.32)}$ | $78.81_{(1.36)}$ | $71.13_{(2.27)}$ | $79.71_{(1.02)}$ | $77.90_{(1.07)}$ | $78.84_{(1.19)}$ | $72.58_{(1.34)}$ | $\mathbf{80.02}_{(0.76)}$ |
| drug (n=40) | $77.71_{(0.32)}$ | $71.86_{(1.07)}$ | $74.30_{(0.59)}$ | $\mathbf{75.79}_{(0.39)}$ | $71.33_{(0.88)}$ | $72.76_{(0.97)}$ | $69.46_{(2.23)}$ | $72.74_{(1.70)}$ | $71.08_{(1.72)}$ | $73.07_{(1.03)}$ | $64.89_{(1.39)}$ | $73.64_{(0.87)}$ | $62.51_{(3.03)}$ | $70.97_{(1.91)}$ | $69.23_{(1.68)}$ | $69.78_{(1.46)}$ | $65.22_{(1.37)}$ | $70.30_{(1.04)}$ |
| covid (n=100) | $74.52_{(0.16)}$ | $71.57_{(0.48)}$ | $73.77_{(0.27)}$ | $\mathbf{74.71}_{(0.34)}$ | $70.71_{(0.46)}$ | $72.76_{(0.44)}$ | $69.05_{(0.96)}$ | $72.13_{(0.66)}$ | $71.60_{(0.59)}$ | $73.22_{(0.46)}$ | $63.52_{(1.29)}$ | $72.04_{(0.57)}$ | $64.25_{(1.51)}$ | $72.64_{(0.64)}$ | $70.08_{(0.67)}$ | $70.78_{(0.59)}$ | $69.05_{(0.48)}$ | $71.96_{(0.49)}$ |
| cutract (n=100) | $72.36_{(0.47)}$ | $70.96_{(0.68)}$ | $70.20_{(0.45)}$ | $\mathbf{72.51}_{(0.55)}$ | $69.97_{(0.97)}$ | $71.94_{(0.89)}$ | $67.94_{(1.01)}$ | $72.42_{(0.66)}$ | $70.53_{(1.51)}$ | $71.98_{(1.39)}$ | $55.72_{(2.04)}$ | $69.14_{(0.93)}$ | $67.59_{(0.71)}$ | $72.42_{(0.54)}$ | $68.79_{(0.83)}$ | $69.68_{(0.76)}$ | $66.89_{(1.03)}$ | $71.52_{(0.70)}$ |
| maggic (n=100) | $67.46_{(0.07)}$ | $59.65_{(0.50)}$ | $58.98_{(0.29)}$ | $\mathbf{61.32}_{(0.42)}$ | $55.71_{(0.83)}$ | $58.90_{(0.72)}$ | $57.20_{(0.91)}$ | $59.34_{(0.64)}$ | $57.26_{(0.50)}$ | $58.28_{(0.46)}$ | $49.54_{(0.71)}$ | $57.91_{(0.74)}$ | $56.36_{(0.54)}$ | $60.11_{(0.54)}$ | $58.89_{(0.51)}$ | $58.99_{(0.42)}$ | $56.17_{(0.68)}$ | $58.86_{(0.69)}$ |
| seer (n=100) | $87.79_{(0.07)}$ | $83.95_{(0.32)}$ | $84.45_{(0.38)}$ | $\mathbf{85.37}_{(0.47)}$ | $83.92_{(0.41)}$ | $85.08_{(0.32)}$ | $81.60_{(0.73)}$ | $85.14_{(0.36)}$ | $83.04_{(0.78)}$ | $84.83_{(0.49)}$ | $70.32_{(2.52)}$ | $83.83_{(0.39)}$ | $81.16_{(1.03)}$ | $85.03_{(0.39)}$ | $81.82_{(0.45)}$ | $82.49_{(0.46)}$ | $78.88_{(0.71)}$ | $84.50_{(0.44)}$ |
| compas (n=100) | $67.18_{(0.30)}$ | $62.56_{(0.72)}$ | $68.02_{(0.29)}$ | $\mathbf{68.19}_{(0.37)}$ | $60.10_{(1.60)}$ | $62.47_{(1.11)}$ | $60.01_{(1.16)}$ | $63.73_{(0.99)}$ | $58.32_{(0.91)}$ | $61.34_{(0.94)}$ | $59.97_{(0.82)}$ | $64.19_{(0.43)}$ | $60.02_{(0.63)}$ | $64.04_{(0.60)}$ | $61.44_{(0.84)}$ | $61.73_{(0.81)}$ | $59.97_{(1.00)}$ | $62.82_{(1.05)}$ |
| adult (n=100) | $84.34_{(0.07)}$ | $81.24_{(0.48)}$ | $46.09_{(1.86)}$ | $74.57_{(1.74)}$ | $47.56_{(3.43)}$ | $73.97_{(1.57)}$ | $74.29_{(1.23)}$ | $80.45_{(0.41)}$ | $75.93_{(1.45)}$ | $78.22_{(1.22)}$ | $77.09_{(0.74)}$ | $\mathbf{81.66}_{(0.53)}$ | $70.70_{(1.02)}$ | $81.04_{(0.37)}$ | $80.56_{(0.53)}$ | $81.10_{(0.44)}$ | $74.04_{(1.44)}$ | $80.23_{(0.98)}$ |
| drug (n=100) | $78.00_{(0.19)}$ | $73.58_{(0.72)}$ | $76.24_{(0.64)}$ | $\mathbf{76.74}_{(0.51)}$ | $69.46_{(2.43)}$ | $71.05_{(2.30)}$ | $68.19_{(2.06)}$ | $73.28_{(1.41)}$ | $72.43_{(0.96)}$ | $73.79_{(0.65)}$ | $67.26_{(1.22)}$ | $75.28_{(0.46)}$ | $62.67_{(2.23)}$ | $73.12_{(0.97)}$ | $70.90_{(1.08)}$ | $71.53_{(1.13)}$ | $68.22_{(1.60)}$ | $73.59_{(0.97)}$ |
| covid (n=200) | $74.69_{(0.19)}$ | $72.33_{(0.52)}$ | $73.40_{(0.25)}$ | $74.62_{(0.13)}$ | $70.70_{(0.73)}$ | $73.12_{(0.48)}$ | $71.07_{(0.35)}$ | $73.89_{(0.40)}$ | $72.47_{(0.50)}$ | $74.44_{(0.44)}$ | $65.55_{(0.67)}$ | $73.07_{(0.38)}$ | $65.04_{(0.69)}$ | $72.90_{(0.37)}$ | $71.68_{(0.49)}$ | $71.87_{(0.49)}$ | $67.89_{(0.56)}$ | $72.38_{(0.42)}$ |
| cutract (n=200) | $72.52_{(0.49)}$ | $71.75_{(0.71)}$ | $71.39_{(0.76)}$ | $73.01_{(0.65)}$ | $70.28_{(0.67)}$ | $72.39_{(0.84)}$ | $69.28_{(0.55)}$ | $72.41_{(0.65)}$ | $71.83_{(0.65)}$ | $\mathbf{74.03}_{(0.69)}$ | $66.66_{(0.94)}$ | $72.49_{(0.71)}$ | $68.77_{(0.71)}$ | $73.16_{(0.68)}$ | $70.23_{(0.83)}$ | $70.80_{(0.79)}$ | $66.61_{(0.76)}$ | $71.87_{(0.71)}$ |
| maggic (n=200) | $67.37_{(0.06)}$ | $61.39_{(0.44)}$ | $58.92_{(0.47)}$ | $\mathbf{61.41}_{(0.39)}$ | $57.33_{(0.51)}$ | $60.16_{(0.41)}$ | $58.48_{(0.59)}$ | $61.33_{(0.57)}$ | $56.26_{(0.42)}$ | $57.20_{(0.91)}$ | $50.74_{(0.74)}$ | $59.60_{(0.55)}$ | $55.95_{(0.72)}$ | $60.75_{(0.39)}$ | $60.73_{(0.44)}$ | $60.78_{(0.49)}$ | $57.18_{(0.56)}$ | $60.23_{(0.44)}$ |
| seer (n=200) | $87.84_{(0.08)}$ | $84.63_{(0.35)}$ | $84.39_{(0.19)}$ | $85.56_{(0.30)}$ | $83.48_{(0.41)}$ | $84.80_{(0.34)}$ | $82.04_{(0.70)}$ | $85.34_{(0.36)}$ | $84.39_{(0.44)}$ | $\mathbf{86.57}_{(0.27)}$ | $82.15_{(0.43)}$ | $86.03_{(0.23)}$ | $77.73_{(1.65)}$ | $85.19_{(0.22)}$ | $83.38_{(0.32)}$ | $84.15_{(0.42)}$ | $79.71_{(0.58)}$ | $85.26_{(0.25)}$ |
| compas (n=200) | $67.14_{(0.20)}$ | $63.27_{(0.60)}$ | $67.02_{(0.53)}$ | $\mathbf{68.15}_{(0.40)}$ | $60.48_{(1.66)}$ | $63.39_{(1.28)}$ | $60.58_{(0.83)}$ | $64.32_{(0.78)}$ | $60.60_{(0.90)}$ | $63.52_{(1.04)}$ | $61.11_{(0.63)}$ | $65.08_{(0.53)}$ | $56.58_{(1.03)}$ | $63.60_{(0.83)}$ | $61.99_{(0.64)}$ | $62.80_{(0.61)}$ | $60.15_{(1.14)}$ | $63.99_{(0.71)}$ |
| adult (n=200) | $84.25_{(0.04)}$ | $82.12_{(0.41)}$ | $40.96_{(2.47)}$ | $75.84_{(1.43)}$ | $49.89_{(3.22)}$ | $76.11_{(1.43)}$ | $78.18_{(0.26)}$ | $82.32_{(0.29)}$ | $81.66_{(0.19)}$ | $83.17_{(0.20)}$ | $80.06_{(0.53)}$ | $\mathbf{83.32}_{(0.35)}$ | $74.31_{(0.81)}$ | $82.64_{(0.33)}$ | $82.26_{(0.38)}$ | $82.39_{(0.33)}$ | $75.21_{(0.69)}$ | $82.02_{(0.25)}$ |
| drug (n=200) | $77.36_{(0.51)}$ | $76.10_{(0.45)}$ | $75.58_{(0.55)}$ | $76.06_{(0.42)}$ | $70.66_{(1.56)}$ | $72.81_{(1.18)}$ | $71.31_{(1.17)}$ | $75.98_{(0.76)}$ | $69.61_{(2.76)}$ | $71.79_{(1.99)}$ | $72.35_{(0.61)}$ | $\mathbf{77.41}_{(0.56)}$ | $65.25_{(2.47)}$ | $75.26_{(0.77)}$ | $74.38_{(0.50)}$ | $74.78_{(0.54)}$ | $68.39_{(1.26)}$ | $74.33_{(0.61)}$ |

## C.3 DECOUPLING PRIOR KNOWLEDGE AND DATA MODEL

Two components can be attributed to the good performances of CLLM: the background knowledge of the LLM, and its capacity to build a strong data model. In this subsection, we provide insights to understand the effect of the LLM's background knowledge (e.g. prior). We considered the Covid dataset (private medical dataset, to avoid memorization issues) and generated data with GPT-4 (same as Section 2.1). We ablate the prompt used in our work (detailed in Appendix B.5), and solely provide one in-context example in the prompt, in order to give the LLM the minimal amount of information about the desired structure of the dataset. This lack of examples forces the LLM to rely on its own prior (background knowledge), and removes the effect of in-context examples which could be used to build a data model. We report the results in Table 11, along with the results for CLLM. From these results, we conclude the following:

1. The LLM prior permits to obtain good downstream performance, but it is outperformed by $\mathcal{D}_{\text{oracle}}$ by a margin of $4.4\%$. Hence, we cannot solely rely on the prior.

2. Downstream performance increases as the number of in-context samples increases. This shows it is indeed important to include the in-context examples if we wish to obtain downstream performance close to $\mathcal{D}_{\text{oracle}}$, as the LLM can build a good data model.

This implies that while the LLM does use background knowledge of similar datasets, it still requires in-context samples to refine its prior by creating a good data model.

Table 11: Downstream accuracy when varying the number of in-context samples in the prompt to generate the augmented datasets.

| In-context samples | Downstream accuracy |
|---|---|
| $n = 1$ (Prior) | $70.20 \pm 1.60$ |
| $n = 20$ | $73.87 \pm 0.50$ |
| $n = 40$ | $73.95 \pm 0.67$ |
| $n = 100$ | $74.71 \pm 0.34$ |
| $D_{\text{oracle}}$ | $74.6 \pm 0.15$ |

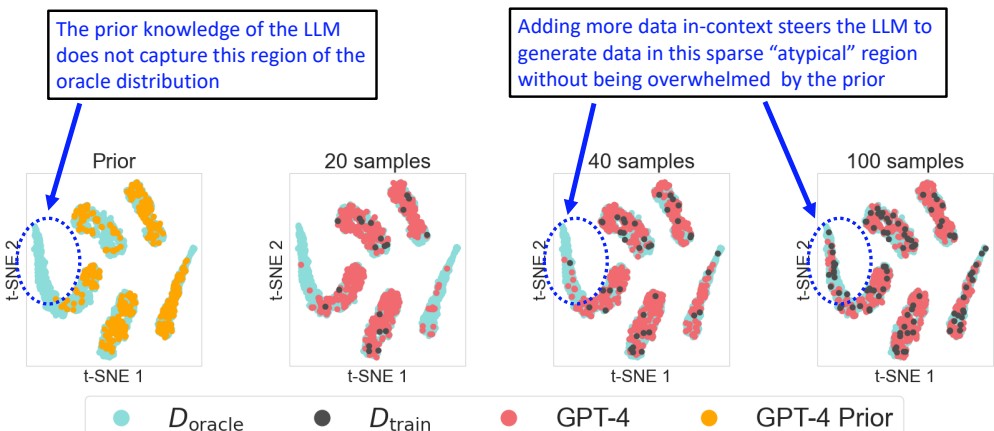

Figure 8: The data generated by the LLM captures the distinct features in "atypical" regions of the Oracle manifold, as in-context samples are added to the prompt. This shows that it is flexible enough to adapt its prior knowledge to the nuances of the data. The group encircled in blue represents patients who are $> 88$ years old, representing around $3.5\%$ of the Oracle. This illustrates the added in-context samples can successfully guide the LLM to generate these rare samples.

Next, we quantify and visualize the strength of the prior, by studying how much the LLMs output distribution adapts to the in-context samples provided. We evaluate data generated by the prior of the LLM ($n = 1$), and for $n = 20, 40, 100$ on the Covid dataset.

In particular, we observe in Figure 8 that there is a region in the oracle data which is not captured by the LLM's prior output (the left part of the leftmost blob, circled in blue in Figure 8). However, as the number of in-context real examples increases in the prompt of the LLM, we observe that this steers the LLM to generate data which covers this region. This region is associated to the subgroup of people older than 87 years old, and having many severe comorbidities (e.g. Diabetes, Cardiovascular diseases) and many respiratory symptoms. This subgroup, in the Oracle dataset, represents less than $3.5\%$ of the data, and is completely ignored by the GPT-4 prior. In particular, the prior defaults to more typical patients in the range 70-80 years old. On the contrary, as $n$ increases, the LLM is guided by the in-context samples and generates samples from this subgroup, which are "rarer" or different from the general population.

This demonstrates that the LLM captures the distinct features of this particular region, and hence is not overwhelmed by the prior, but instead the data in the form of in-context samples adapts it, hence aligning the augmented dataset with the ground-truth distribution.

## C.4    ABLATION FOR CONTEXTUAL INFORMATION ON COMPAS

We conduct a similar experiment as in Table 2, and use the dataset Compas. We report the results in Table 12.

Table 12: Including contextual information in the prompt improves precision (P), recall (R), and utility (U) in low-sample settings (results shown for Compas).

| $n_{samples}$ in $D_{train}$ | GPT-4 w/ context | | | GPT-4 no context | | | TVAE | | |
|---|---|---|---|---|---|---|---|---|---|
| | P | R | U | P | R | U | P | R | U |
| 20 | $\mathbf{0.69}_{(0.02)}$ | $\mathbf{0.88}_{(0.02)}$ | $\mathbf{0.69}_{(0.02)}$ | $0.27_{(0.03)}$ | $\mathbf{0.89}_{(0.03)}$ | $0.60_{(0.03)}$ | $0.43_{(0.02)}$ | $0.43_{(0.05)}$ | $0.55_{(0.04)}$ |
| 40 | $\mathbf{0.70}_{(0.0)}$ | $\mathbf{0.92}_{(0.01)}$ | $\mathbf{0.65}_{(0.03)}$ | $0.31_{(0.06)}$ | $0.84_{(0.03)}$ | $0.57_{(0.01)}$ | $0.54_{(0.02)}$ | $0.80_{(0.02)}$ | $0.50_{(0.04)}$ |
| 100 | $\mathbf{0.69}_{(0.02)}$ | $\mathbf{0.89}_{(0.02)}$ | $\mathbf{0.69}_{(0.01)}$ | $0.34_{(0.1)}$ | $0.85_{(0.05)}$ | $0.62_{(0.01)}$ | $0.60_{(0.03)}$ | $0.86_{(0.02)}$ | $0.59_{(0.03)}$ |
| 200 | $\mathbf{0.70}_{(0.01)}$ | $\mathbf{0.89}_{(0.02)}$ | $\mathbf{0.69}_{(0.01)}$ | $0.31_{(0.05)}$ | $0.87_{(0.03)}$ | $0.58_{(0.05)}$ | $0.65_{(0.02)}$ | $0.88_{(0.01)}$ | $0.63_{(0.01)}$ |

These results highlight the importance of incorporating contextual information in the prompt, as it enables to exploit the prior knowledge of the LLM.

## C.5 Comparison to random noise baseline

We now compare to a random noise baseline. Specifically, where we augment the dataset with random additive Gaussian noise. In order to capture the correlations between the different features, we fit a Kernel Density Estimator with a Gaussian kernel and bandwidth given by Scott's rule. We then sample 1000 points to create an augmented dataset $D_{\text{syn}}$. We report the performance gap

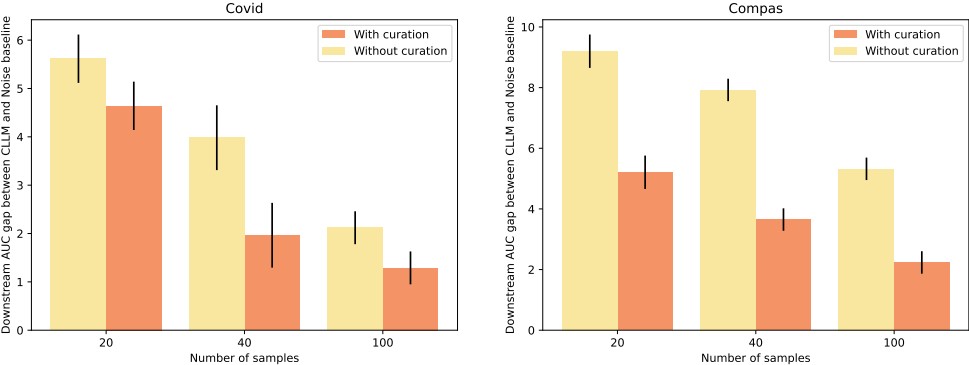

Figure 9: The random noise baseline does not match the performance of CLLM

between CLLM and this baseline (with and without curation) for the Covid and Compas datasets in Figure 9.

We observe that the random noise baseline does not match the performance of CLLM (i.e. has a performance gap), although the baseline naturally improves as the dataset $D_{\text{train}}$ grows in size.