# OpenReview forum: "Curated LLM: Synergy of LLMs and Data Curation for tabular augmentation in ultra low-data regimes"
_ICLR.cc/2024/Conference — Submitted to ICLR 2024_

### Official Review · Reviewer_4AgE · 2023-10-25

**Soundness:** 1 poor
**Presentation:** 3 good
**Contribution:** 2 fair
**Rating:** 6
**Confidence:** 4

**Summary:**

This paper investigates the problem of improving ML performance on ultra small tabular datasets via synthetic data augmentation. The authors introduced a method that (1) generates new data by using the small training set as the context and querying an LLM, and then (2) further filters the synthesized data by looking at the learning dynamics of the synthetic data. The authors compare the proposed method to existing methods of data synthesis and augmentation, and show the superiority of the proposed method in the ultra-low data regime (n < 100).

**Strengths:**

- The topic on data synthesis for ultra-small dataset is interesting and has far-reaching consequences.
- The concepts touched upon along the way, specifically data prior and data quality flags, offer a framework to think about this problem.
- The writing is clear, and the paper is well organized.

**Weaknesses:**

- The effect and origin of the LLM’s prior are unclear, so it is difficult to confidently attribute the performance gain to in-context learning.
- The curation method seems to be model-dependent and may intensify the weaknesses of the ML model.
- Because the method proposed is quite simple to use, I feel we need to know when the methods will break down in order to understand its pitfalls. There is not really anything on that in the paper.

**On the prior:**
Section 2.1 suggests that the LLM has a strong prior, as it is able to extrapolate synthesis to regions without any training data. The question that immediately follows is whether the in-context learning is flexible enough to learn the nuances in the ultra-small dataset, or will the LLM’s prior overwhelm those nuances? Note that Table 3 shows that performance does not improve much as n increases for LLM models relative to  non-LLM models. This observation is consistent with the prior being too strong. Of course, it is also consistent with the in-context learning being very good. I hope the authors could tease these two scenarios apart.

Another question relates to the scenario of “truly” scarce data, i.e., data that is really not in the training data of the LLM. Although the authors mention that 4 of the datasets are unlikely to be in the training of the LLM due to their being closed-source, we will need a way to quantify that likelihood to get at the answer. In fact, Table 3 shows that for n=20, most of the non-LLM  Uncur methods have worse utility than D_train, whereas most of the LLM Uncur methods have higher utility than D_train. This observation is consistent with LLM having memory of the data. Of course, this is also consistent with good in-context few-shot learning. It will be important to distinguish between memorization and in-context few-shot learning for the truly scarce scenario.

For both questions above, one idea to get at them is to create datasets that are definitely not in the training of the LLM by manually manipulating the column dependencies and marginal distributions of existing data, then see if the results still hold.

**On the curation:**
It seems that the curation depends on the model via the f_e(x) (see equations 1 and 2), therefore, the augmentation is optimized for a model and can make the model’s specific inductive biases more pronounced. Some thoughts related to addressing this include: (1) curating based on a ML model and evaluating on many models, (2) include more metrics such as resemblance. I hope the authors could provide a clear rationale for addressing this issue.

The selection criteria based on high aleatoric uncertainty makes it similar to active learning. This means that the curation will make the model perform better even from random sampling, as evidenced by curation increasing performance for all the methods. It may be worth it to make this connection.

Looking at the two criteria, do they not have a 1-to-1 relationship, as [f_e(x)]_y is the only variable in both quantities? That is, are the two criteria equivalent to selecting a range of confidence or a range of uncertainty? Maybe worth clarifying.

**On pitfalls:**
As mentioned, demonstration of pitfalls are extra important because the method proposed is very easy to use. Such demonstration will likely also shed light on the previous questions.

**Questions:**

**Questions and comments:**
- Does the LLM handle continuous variables well?
- Figure 2: it’s hard to read with solid markers occluding each other.
- Table 1: what quantity is the performance number?
- Figure 3: surprisingly smooth, and the dip around 6 samples stands out. What’s the explanation? What is the actual performance after augmentation — want to check if the gain is substantial?
- Don’t think ICL is ever introduced as an acronym.

**Details Of Ethics Concerns:**

The method proposed is very easy to apply, and the paper is framed in terms of using it in fields like medicine. Without a good understanding of the adverse effects of the method, it may unknowingly cause problems.

---

> ### Author Response · Authors · 2023-11-19
> **Response to Reviewer 4AgE [Part 1/5]**
>
> Dear Reviewer 4AgE
>
> Thank you for your thoughtful comments and suggestions! We give answers to each of the following points in turn and highlight the updates to the revised manuscript. In addition, we have uploaded the revised manuscript. We hope this response alleviates your concerns, but please let us know if there are any remaining concerns.
>
> - (A) Identifying possible pitfalls of CLLM **[Part 2/5]**
> - (B) Priors of LLMs for data augmentation **[Part 3/5]**
> - (C) Details on data curation **[Part 4/5]**
> - (D) Answers to the other questions **[Part 5/5]**

---

> ### Author Response · Authors · 2023-11-19
> **Response to Reviewer 4AgE [Part 2/5]**
>
> ### (A) Identifying possible pitfalls of CLLM
> Thank you for raising the point about the possible pitfalls of LLMs. We agree that there are risks when using large language models (LLMs), as explained in Sec. 1 and Sec. 2.2 in our manuscript, especially for sensitive applications like healthcare and finance. This motivated our contribution of post-generation curation as a step towards addressing this issue. That said, we agree more should be done to discuss these challenges of using LLMs in the paper and possible "pitfalls".
>
> We wish to highlight our *'Hardness' Proxy Signal* introduced in **Section 3.2**, which is a way to identify failure cases of data augmentation (i.e. proactively flagging the "pitfall").
>
> In this part of the paper, we highlight that, given the black-box nature of an LLM, we might not trust generated datasets and specifically those that will lead to poor downstream performance.  The inclusion of our 'hardness' proxy serves as a proactive measure to flag potentially problematic datasets reflecting our commitment to ensuring the quality and reliability of the data used in our models. To build this proxy signal, we directly leverage the output of the curation module, with the rationale stated in _Section 3.2_: $D_{syn}$ should intuitively be considered imperfect if curation discards many of its samples since the number of discarded samples measures the quality of samples with respect to the small but goldstandard $D_{train}$.
>
> This proxy signal aligns with the results in Table 3, where we show that CLLM struggles with the Adult dataset before curation. **Figure 6** shows that the proportion of discarded samples is the highest for this dataset.
>
> Consequently, this shows a practitioner _could_ have used this proxy signal to flag this failure case of the LLM before using it to train a downstream predictive model.
>
> We hope this clarifies how we proactively identify pitfalls and failure cases from the augmentation.
>
>
> **Limitations and warning readers.**
> Of course, we note that CLLM is only a first step towards better synthetic data. In practice, we thus agree that the output of generative models, including CLLM, should always be exhaustively evaluated on quality, fairness, and other forms of bias, before application to sensitive areas like healthcare, to avoid any potentially harmful applications.
>
> We have consulted an academic ethicist to discuss how these ethical concerns should be best communicated. To ensure readers may not inadvertently cause bias and harm in the real world through applying CLLM, we have included the following warning at the end of the introduction:
>
> *LLMs may make errors and may reflect or exacerbate societal biases that are present in their data [R1]. Though the curation in CLLM improves synthetic data quality, it does not directly aim to remove biases. The quality and fairness of generated data should always be evaluated. More research into LLM bias is required before methods like CLLM should be applied to real-world sensitive settings like healthcare and finance.*
>
> Furthermore, we extended the Ethics statement to include this warning and will include the same warning in the code repository.
>
> Thank you again for bringing up this important issue and we hope that our clarifications have addressed your concerns.
>
> **UPDATE:** we have updated Section 1 and our Ethics statement in our revised manuscript with a warning.

---

> ### Author Response · Authors · 2023-11-19
> **Response to Reviewer 4AgE [Part 3/5]**
>
> ### (B) Priors of LLMs for data augmentation
> We would like to thank the Reviewer for bringing this interesting point on the prior modeled by the LLM. We agree with the Reviewer that two components can be attributed to the good performances of CLLMs: the prior given by the LLM, and the in-context examples present in the prompt.
>
> The reviewer also rightfully points to possible memorization as the source of performance. We highlight that this aspect of memorization motivated the choice of datasets in our manuscript, with $4$ private datasets. These "medical datasets" are proprietary datasets which not in the public domain and hence we have ensured they are not accessible as part of the GPT-4 or GPT-3.5 training corpora which would have originated from web scraping or "actual release" of these datasets. Hence, we cannot attribute strong performance on these datasets to memorization but rather due to background knowledge on "similar data" or via the in-context examples.
>
> We thank the Reviewer for the suggestion regarding the prior. We have conducted a new experiment to understand the effect of the LLMs background knowledge (e.g. prior).
>
> We considered the Covid dataset (private medical dataset, to avoid memorization issues) and generated data with GPT-4 (same as Sec 2.1). We ablate the prompt used in our work (detailed in Appendix B.5), and solely provide one in-context example in the prompt, in order to give the LLM the *minimal* amount of information about the desired structure of the dataset.
>
> This lack of examples forces the LLM to rely on its own prior (background knowledge), and removes the effect of in-context examples which could be used to build a data model.
>
> We report the results for CLLM in the following table:
> | In-context samples |  Downstream accuracy  |
> |--------------------|----------------|
> | n=1 (Prior)        | 70.20+-1.60    |
> | n=20               | 73.87+-0.50    |
> | n=40               | 73.95+-0.67    |
> | n=100              | 74.71+-0.34    |
> | $D_{\mathrm{oracle}}$| 74.6 +- 0.15 |
>
>
>
> From these results, we conclude the following:
>
> (1) The LLM prior permits to obtain good downstream performance, but it is outperformed by $\mathcal{D_{oracle}}$ by a margin of $4.4\%$. Hence, we cannot solely rely on the prior.
>
> (2) Downstream performance increases as the number of in-context samples increases. This shows it is indeed important to include the in-context examples if we wish to obtain downstream performance close to $\mathcal{D_{oracle}}$, as the LLM can build a good data model.
>
> This implies that while the LLM does use background knowledge of similar datasets, it still requires in-context samples to refine its prior by creating a good data model.
>
> **UPDATE**: We included these results in Appendix C.3 of our updated manuscript

---

> ### Author Response · Authors · 2023-11-19
> **Response to Reviewer 4AgE [Part 4/5]**
>
> ### (C) Details on data curation
>
> We provide answers to three aspects of data curation: (i) Curation model, (ii) Relation to active learning and (iii) Clarifying difference of criteria/metrics
>
>
> **(i) Curation model**
>
> We thank the reviewer for the point on curation and the model used to perform it. The curation indeed requires training a model to compute the metrics of confidence and aleatoric uncertainty.
>
> The suggestion of the reviewer ("curating based on an ML model and evaluating on many models") is an approach that we have adopted in our submitted manuscript, and the detailed results are given in **Appendix C.1**. We apologize if this was unclear.
>
> The curation, on the one hand, was performed by training an XGBoost model, leading to a curated dataset.
>
> On the other hand, we evaluated the downstream performance with $4$ different types of classifiers namely: XGBoost, Random forest, Logistic regression and Decision Tree, which were trained on the curated dataset.
>
> As we can see, the conclusion that curation helps improve downstream performance holds for each these various downstream models, as is indicated by the green arrows in Tables 6,7,8,9.
>
> **Update:** We have made this point clearer by adding this discussion in Appendix C.
>
> **(ii) Relation to Active Learning**
> We also thank the reviewer for bringing up this interesting point about active learning and our curation step. While we have already discussed some differences to active learning in the Extended related work section (Appendix A), we now provide a detailed discussion on how they differ on the uncertainty metrics they use.
>
> Active learning primarily focuses on the iterative process of selecting data samples that, when labeled, are expected to most significantly improve the model's performance. This selection is typically based on criteria such as uncertainty sampling which focuses on **epistemic uncertainty** [R1-R4]. The primary objective is to minimize labeling effort while maximizing the model's learning efficiency. Additionally, active learning would aims to label instances based on epistemic uncertainty where the model struggles to make accurate predictions, yet the samples themselves are _correct_.
>
> In contrast, CLLMs leverage training dynamics based on **aleatoric uncertainty** and confidence and is designed to discard samples that might jeopardize the downstream accuracy. These samples can be considered to have inherent issues or are erroneous, such as being "mislabeled".
>
> To summarize, in active learning, epistemic uncertainty is used to identify data points that, if labeled, would yield the most significant insights for model training. In our approach, they serve to identify and exclude/filter data points that could potentially deteriorate the model's performance.
>
> **UPDATE:** We have included this discussion on uncertainty metrics in Appendix A in our revised manuscript.
>
> **(iii) Clarifying difference of criteria/metrics**
>
> Finally, let us explain why confidence and the aleatoric uncertainty do not have a 1-to-1 relationship.
>
>  Let us consider a setting with $e \in$ {1,2}, and
>
> $[f_{1}(x)]_{y}=0.1$ and
>
> $[f_{2}(x)]_{y}=0.9$
>
> Then the confidence is equal to $\frac{0.1 + 0.9}{2} = 0.5$ and the aleatoric uncertainty is equal to $\frac{0.1 * 0.9 + 0.1 * 0.9}{2} = 0.09$.
>
> Consider an alternative case where:
>
> $[f_{1}(x)]_{y}= 0.5$ and
>
> $[f_{2}(x)]_{y}=0.5$
>
> The confidence is also equal to $0.5$ but the aleatoric uncertainty is equal to $\frac{0.5 * 0.5+0.5 * 0.5}{2} = 0.25$. Hence these two examples share the same confidence but differ in their aleatoric uncertainty, which demonstrates why these two notions do not capture the same aspect of the learning dynamics.

---

> > ### Author Response · Authors · 2023-11-19
> > **Response to Reviewer 4AgE [Part 5/5]**
> >
> > ### (D) Answers to the other questions
> > - LLMs handling continuous variables
> >
> > Thanks for the question. Yes, the LLMs do handle continuous variables. Our datasets contain a mixture of continuous and categorical variables. For example, variables such as age or biomarkers which are continuous. The LLMs capability to generate samples resulting in good downstream model performance is hence indicative of this ability to handle such continuous variables.
> >
> > - Figure 2 markers
> >
> > The reason for putting the markers over one another is to show how each of the different augmentation approaches matches the oracle (in turquoise). We apologize if this was unclear that the Oracle fills all the space of the regions shown and that we plot the other markers (GPT-4, $\mathcal{D_train}$, TVAE) and on top of it.  We have updated the caption to clarify this
> >
> > **UPDATE**: We updated the caption of Figure 2 to clarify that the markers should ideally match the underlying oracle.
> >
> > - Table 1: performance measure
> >
> > We report in Table 1 the average gain in downstream accuracy compared to the downstream accuracy of a model trained on $\mathcal{D_train}$.
> >
> > **UPDATE**: We updated the column in Table 1 to clarify this in our revised manuscript.
> >
> >
> > - Figure 3 Clarification and gain.
> >
> > We clarify how the curve in Figure 3 is obtained. We stratified the list of subgroups depending on their number of samples and then took the average of the performance gain using a model trained on CLLM data (i.e. augmentation) versus a model trained on $\mathcal{D_train}$. Naturally, not all points on the x-axis have the same number of subgroups, and this variability explains the slight dip at $6$ samples. The goal of the figure is to highlight that the performance gains from augmentation are largest for subgroups that originally had the fewest samples in $\mathcal{D_train}$. Hence, the result highlights that augmentation improves downstream performance on small subgroups. Finally, as suggested by the Reviewer, we clarify the actual performance gain (overall) after augmentation: augmentation gives an AUC gain of 5.28%.
> >
> > - ICL acronym definition
> >
> > We thank the reviewer and have defined the acronym ICL (in-context learning) in our revised manuscript.
> >
> > **UPDATE**: Updated the manuscript to define the acronym in Section 2.
> >
> >
> > We hope this answers your points, please let us know if there are any remaining concerns.
> >
> > ----
> > -----
> >
> > ### References
> > [R1] Stephen Mussmann and Percy Liang. On the relationship between data efficiency and error for uncertainty sampling, 35th International Conference on Machine Learning, PMLR.
> >
> > [R2]Neil Houlsby, Ferenc Huszar, Zoubin Ghahramani, and Mate Lengyel. Bayesian active learning for classification and preference learning. arXiv preprint arXiv:1112.5745, 2011.
> >
> > [R3] Andreas Kirsch, Joost van Amersfoort, and Yarin Gal. Batchbald: Efficient and diverse batch acquisition for deep bayesian active learning. In Advances in Neural Information Processing Systems, pp. 7024–7035, 2019.
> >
> > [R4] Nguyen, Vu-Linh, Mohammad Hossein Shaker, and Eyke Hüllermeier. “How to measure uncertainty in uncertainty sampling for active learning.” Machine Learning 111, no. 1 (2022): 89-122.

---

> > > ### Comment · Reviewer_4AgE · 2023-11-20
> > >
> > > Thank the authors for their response. I especially appreciate the additional experiment with n=1 example and the explanation on differences to active learning. Having said that, I feel most of my concerns are only touched upon but not fully addressed. I still have little knowledge about what the limitation and breaking points of CLLM are. The n=1 results further confirm that the prior of the LLM is overwhelmingly strong and learning to move away from the prior may require MANY examples. Lastly, for the curation, what happens if the curation is done using Logistic regression (instead of XGBosst), and then tested on XGBoost, Random forest, Logistic regression and Decision Tree? In general, I am hoping the authors would demonstrate that the curation model does not taint the curated data with its weaknesses. Using Logistic regression (a linear, less expressive model than XGBoost) as a curation model will better convince me.

---

> > > > ### Author Response · Authors · 2023-11-20
> > > > **Thank you for your response**
> > > >
> > > > Dear Reviewer 4AgE,
> > > >
> > > > We would like to thank you for your response.
> > > >
> > > > We address the issues around (i) prior knowledge and (ii) possible pitfalls in the choice of the curation model.
> > > >
> > > > **(i) prior knowledge**
> > > >
> > > > On the point of prior knowledge, we want to stress that having substantial prior knowledge is a desirable property for the LLM in our ultra-low data regime and a motivation for CLLM. Indeed, the absence of such knowledge would make inferring the underlying data structure from just a few examples infeasible.
> > > > That is why we highlight in **Section 2.1** in our manuscript that prior knowledge of similar problems is a desirable property, especially in regimes where we have limited real data. Hence, this capability sets LLMs apart from conventional tabular synthetic generators. With that in mind, as shown in our ablation in Appendix C.3, the "prior" is not sufficient to match performance on the Oracle.
> > > >
> > > > We want to highlight that while adapting this prior requires access to a certain number of samples, this number is actually _very small_ compared to the size of the $D_{\mathrm{oracle}}$.  For example, $n=100$ represents only 3% of the full data size, while enabling CLLM to adapt the prior knowledge to the nuances of the dataset, with a jump of $4.5%$ in accuracy. This puts it on par with the downstream performance obtained by training a model on  $D_{\mathrm{oracle}}$. $n=100$ should also be compared to the dimensionality of the dataset, with $d = 29$ for the Covid dataset for example. These points show the challenges of the ultra-low data regime that CLLM addresses.
> > > >
> > > >
> > > > **(ii) possible pitfall in the choice of the curation model**
> > > >
> > > > We would like to thank the Reviewer for the suggestion of using a Logistic Regression model to perform curation. We have conducted this experiment, and report the results in the Table below, when the curation is performed by training a logistic regression model, for the datasets Covid and Compas.
> > > >
> > > >
> > > >
> > > > * Uncur = Uncurated, Cur = Curated by LR
> > > >
> > > > _Covid dataset:_
> > > > | n   | XGB (Uncur/Cur) | RF (Uncur/Cur) | LR (Uncur/Cur) | DT (Uncur/Cur) |
> > > > |-----|-----|----|----|----|
> > > > | 20  | **75.13** / 73.84 | **75.73** / 75.10 | **76.35** / 74.44 | **67.80** / 66.37 |
> > > > | 40  | 74.74 / **75.21** | **75.89** / 75.79 | **76.40** / 75.04 | **66.77** / 66.48 |
> > > > | 100 | 74.86 / **75.49** | 75.69 / **75.98** | 76.40 / **76.79** | 67.00 / **67.19** |
> > > > | 200 | 74.49 / **76.06** | 75.28 / **75.64** | 76.09 / **77.10** | 66.70 / **68.01** |
> > > >
> > > >
> > > > _Compas dataset:_
> > > >
> > > > | n   | XGB (Uncur/Cur) | RF (Uncur/Cur) | LR (Uncur/Cur) | DT (Uncur/Cur) |
> > > > |-----|-----|----|----|----|
> > > > | 20  | **68.43** / 67.31 | **67.96** / 67.23 | **71.45** / 69.19 | **61.85** / 61.20 |
> > > > | 40  | **67.39** / 67.21 | **67.38** / 66.62 | **70.72** / 69.05 | **60.77** / 59.67 |
> > > > | 100 | 68.40 / **69.04** | 68.22 / **68.39** | 71.11 / **70.96** | 62.88 / **63.07** |
> > > > | 200 | 68.09 / **67.86** | **67.84** / 67.24 | 70.63 / **70.66** | 60.76 / **60.96** |
> > > >
> > > >
> > > >
> > > >
> > > >
> > > >
> > > > As we can see, for very few samples ($n=20, 40$), using a logistic regression model does not lead to improved downstream performance. This is actually expected behaviour in the supervised learning setting and is not necessarily unique to our curation mechanism. The performance and capabilities are of course dependent on both the data and the model. Since the data is fixed, it is reasonable to expect the model to play a significant role.
> > > >
> > > > The key problem is the capacity and generalization properties of the curation model. The logistic regression model is only able to output linear decision boundaries, which might not generalize well with very few training samples. This limitation impairs the purpose of curation, which is to model the ground-truth feature-label relationship in order to be able to detect noise in the augmented dataset $\mathcal{D}_{\mathrm{syn}}$.
> > > >
> > > > We highlight that this exemplifies a “pitfall” or failure case of CLLM requested by the reviewer. We cannot expect good performance by choosing “any” curation model, rather we require a curation model with enough capacity and generalization properties  — where boosting methods like XGBoost used in our paper have shown to achieve best performance on tabular data. Consequently, it explains why we achieve best performance with this choice, from a curation perspective.
> > > >
> > > > As such, we give a guideline for the curation step: the model used for curation should be at least as flexible as the model that the practitioner intends to use for the downstream task. This motivates our choice of using an XGBoost model as the curation model in the experiments in our manuscript.
> > > >
> > > > ---
> > > > *We thank the Reviewer for your help in improving our work. Please let us know if our latest changes have addressed your concerns and if there is anything else you would like to see.*

---

> > > > > ### Comment · Reviewer_4AgE · 2023-11-21
> > > > >
> > > > > Thank the authors' quick response and extra experiment! The result confirms the model-dependent effect of the curation method. I hope this limitation would be mentioned in the paper.
> > > > >
> > > > > My remaining concern is about the overwhelming prior. I totally understand that the strong prior is important for the ultra-low regime. But the strong prior is a double-edge sword. Perhaps the downside I am thinking about is clearest illustrated by an example. Suppose covid effects for a particular population is distinct to that for the general population. Suppose further that the LLM DOES NOT know about this particular population because it is truly rare. When we use CLLM to obtain more data, my worry is that ICL query would just generate data for the general population but fails to capture the distinct features of the particular population. In this case, the nuances in this particular population would be overwhelmed by the LLM's prior and harder to learn. This is the kind of situation I hope the authors could directly address.

---

> ### Author Response · Authors · 2023-11-21
> **Thank you for your response & new experiment**
>
> Dear Reviewer 4AgE
>
>
> Thank you for your response!
>
> As you suggested, we have **updated our manuscript** with the discussion on the choice of the curation model, both in Appendix B.3 and in Section 2.2 in our revised manuscript.
>
> We agree that the prior encoded in the LLM can be a double-edged sword, but we also have reasons to be optimistic. First, a realistic prior is better than no prior at all, which is the status quo in deep generative modeling for tabular data. Second, the LLM’s prior may not be as strong as you suggest. We have conducted a new experiment to quantify and visualize the strength of the prior, by studying how much the LLM's output distribution adapts to the in-context samples we provide.  More specifically, we show t-SNE plots for data generated by the prior of the LLM ($n=1$), and for added in-context examples of $n=20,40,100$ on the Covid dataset.
>
> The plots are included in the updated manuscript (Appendix C.3) and at the following link:
>
> **https://i.imgur.com/x00toXC.png**
>
>
> **Interpretation:**
> In particular, we observe a region in the oracle data that is _not_ captured by the LLM’s prior output (the left part of the leftmost blob, circled in blue in our plot). However, as the number of in-context examples (real data) increases in the prompt of the LLM, we observe that the LLM is steered to generate data that covers this region.
>
> We investigated this region and noticed that it is associated with the subgroup of people older than $87$ years old, and having many severe comorbidities (e.g. Diabetes, Cardiovascular diseases) and respiratory symptoms. This subgroup, in the Oracle dataset, represents less than $3.5\%$ of the data, and is completely ignored by the GPT-4 prior. In particular, the prior defaults to more typical old patients in the range of $70$-$80$ years old.
>
> On the contrary,  as $n$ increases, the LLM is guided by the in-context samples and generates samples from this subgroup, which as you mention are "rarer" or different from the general population.
>
> This demonstrates that the LLM captures the distinct features of this particular region, and hence is _not overwhelmed by the prior_, but instead, the data in the form of in-context samples adapts it, hence aligning the augmented dataset with the ground-truth distribution.
>
>
> Additionally, similar to the main paper, we assess the synthetic data using the Precision (Quality) and recall (Diversity) metrics [R1] with respect to $D_{\mathrm{oracle}}$:
>
> | $n_{\mathrm{samples}}$ in $\mathcal{D_{train}}$ | Precision |   Recall |
> | ------------------ | ------------------- | ------------------- |
> | Prior ($n_{\mathrm{samples}}=1$) |   0.29 +-  0.026   |   0.64 +- 0.09        |
> | 20                                 | 0.41+-0.04      | 0.87+-0.03      | 0.74+-0.01     | 0.13+-0.0          | 0.82+-0.01         | 0.66+-0.01         | 0.33+-0.07 | 0.50+-0.03 | 0.59+-0.02 |
> | 40                                 | 0.40+-0.01      | 0.91+-0.01      |
> | 100                                 | 0.42+-0.01      | 0.86+-0.02          |
>
> These results further validate the adaptation of the LLM to capture the nuances of the data, as more in-context samples are provided.
>
>
> **UPDATE:** We have updated Appendix C.3 which now dissects the role of the prior along two dimensions:
>
> (1) The prior alone cannot match the oracle downstream performance and needs in-context samples [``last update``]
>
> (2) The in-context samples guide the LLM generation and is not overwhelmed by the prior --- allowing us to generate rarer subgroups. [``new update``]
>
> ----
>
> _We thank the reviewer for encouraging us to disentangle this important aspect, which has helped us improve the paper!_
>
> _We hope that our clarifications and new experiment have addressed your concerns, please let us know if there is anything else we could do before the discussion period ends._
>
>
> Paper 7282 Authors
>
> ----
>
> [R1] Mehdi SM Sajjadi, Olivier Bachem, Mario Lucic, Olivier Bousquet, and Sylvain Gelly. Assessing generative models via precision and recall. Advances in neural information processing systems, 31,2018.

---

> > ### Comment · Reviewer_4AgE · 2023-11-22
> >
> > Many thanks to the authors for the direct and insightful response. I feel my main concerns have been addressed adequately, and I am happy to raise my score to 6.
> >
> > I do want to stress that part of me is still cautious: The method presented in this paper is very simple and seems effective; hence, it could see quick and wide spread. It is from this perspective that I would really like to know all the conditions under which the approach might produce adverse effects.

---

> ### Author Response · Authors · 2023-11-22
> **Thank you for the response & score increase**
>
> Dear ``Reviewer 4AgE``
>
> Thank you for your response and for raising your score. Your suggestions have greatly helped us improve the paper!
>
>
> We agree that CLLM is only a first step towards better synthetic data. In practice, we thus agree that the output of generative models, including CLLM, should always be exhaustively evaluated on quality, fairness, and other forms of bias, before application to sensitive areas like healthcare, to avoid any potentially harmful applications.
>
> As mentioned during the discussion period, and based on the reviewer's suggestion, we included the following warning at the end of the introduction to ensure readers may not inadvertently cause bias and harm in the real world by applying CLLM:
>
> >"LLMs may make errors and may reflect or exacerbate societal biases that are present in their data [R1]. Though the curation in CLLM improves synthetic data quality, it does not directly aim to remove biases. The quality and fairness of generated data should always be evaluated. More research into LLM bias is required before methods like CLLM should be applied to real-world sensitive settings like healthcare and finance."
>
> Furthermore, we extended the Ethics statement to include this warning and will include the same warning in the code repository.
>
> We are optimistic that CLLM will spur further research, including improving the curation mechanism (e.g. for algorithmic fairness or privacy) and developing proxy signals for failure conditions (such as the Hardness proxy in Section 3.2).
>
> ---
>
> _Thank you again for bringing up this important issue and thank you again for your time and feedback on the paper!_

---

### Official Review · Reviewer_Nkcr · 2023-10-31

**Soundness:** 2 fair
**Presentation:** 3 good
**Contribution:** 2 fair
**Rating:** 5
**Confidence:** 3

**Summary:**

This paper investigates the problem of using LLMs to generate tabluar data for medical applications where labeled data are extremely scarse. Specifically, this work considers the "ultra-low resource" scenario where the task data is <100 samples. This work proposes to use these samples as example with context information to prompt LLMs to generate synthetic data. The work claims the model is able to leverage its medical and other knowledge to generate usable tabular data. Then, the generated data goes through a data curation process, where the data is examined at multiple checkpoint during the model training process. By throwing out data that is inconsistent to the model or nearby samples, the work shows that the curated synthetic data can help improve model performance.

**Strengths:**

The problem being investigated is clearly of interest. Data generation for low-resource scenarios is highly relevant and could be particularly helpful for disadvantaged or marginal groups or rare scenarios. Data generation and curation are also crucial for improving the fairness of ML/AI systems. The potential of foundation models is especially promising.

The language is very clear. The paper is beautifully formatted. It is a comfortable read. The paper is well-motivated and well-contextualized.

The research need is valid and important and the proposed approach is relevant and timely. The idea of post-generation curation is smart. Related work is well introduced and comparisons are thorough

**Weaknesses:**

My major concerns are

1. I don't see the technical contribution of this work;

- This essentially comes down to the few-shot generation problem, where there is much more existing work beyond tabular data. If it is a traditional NLP task such as generating data for sentiment analysis, there are already abundant success cases. Whether the data supports training a model depends on what model you are training and what you want to achieve.

- With less than 100 data but 30 attributes, it is impossible for LLMs to effectively infer the pattern of data. And I doubt the generation is trivial. I wonder how the proposed generation process compares to just adding random noise to augment existing samples before curation. D_train in many cases is very high, while the baseline performance is significantly worse. This raises doubt on whether the setup is correct. I suspect using small random noise to augment the dataset would perform better than many baselines.

2. The proposed approach has a significant overlook for the risk of biases and untrustworthiness of LLMs. The proposed methods for medical applications pose major ethical concerns.

- I strongly doubt the validity of this approach in medical applications. Specifically, what is the rationale for the generated data to be considered "correct" or incorporate the right knowledge or information? **If the LLM's output suggests a high correlation between certain marginal groups and a high prevalence of sexually transmitted diseases (STDs), how do you tell whether this is based on medical publications or social biases? Factual tracing for LLMs is currently known as a very hard problem. Active research is ongoing and there are not yet effective ways to relate the model's output to its training samples.**  Without due effort investigating this issue, using data generated by black-box models as the foundation for building medical applications is irresponsible and poses ethical concerns. Given that the work targets ultra-low resource scenarios, it is especially alerting to associate the risk with historically disadvantaged or marginal groups.

- Without specific treatments, LLMs are generally quite poor in logical/mathematical reasoning. It is often challenging for these models to identify the simplest patterns in input data, which is consistent with the reported no-context generation scenario. For the generation example provided in the appendix, I found it rather concerning.

- GPT models are trained overwhelmingly on web text, which contains a high level of subjective arguments, biases, and ungrounded claims especially for major social topics such as COVID or COVID patients. It could easily incorporate bias between demographic attributes and medical conditions and outcomes. The most concerning part is I don't see discussions on it at all, which makes me worry that the authors may not be fully aware of the tool they are leveraging. Given the seriousness of medical applications, this level of overlook is worrisome.

**Questions:**

- There are existing notions such as "low-resource" refers to less than 5k annotated samples and "strict few-shot" refers to <=16 labeled samples per class. I'm not aware of the definition of "ultra-low-resource". Based on the illustration of this paper, it seems to be < 100 samples. The title suggests ultra-low, but without a definition for it, the abstract and introduction talk about "low-resources". **What is ultra-low? Is this an existing notion or is it proposed by this work?** This range of samples seems to be the case that is often considered "few-shot".

- To have diversity in the generated samples, I would expect the researchers to look into decoding strategies (such as temperature or sampling). It usually does not work by just "asking the model to generate diverse samples".

- Table 3, in "adult" row in the first section, the best performer are not marked.

- It is better to provide average performance (with standard deviation) for each method for easy comparison between methods.

**Details Of Ethics Concerns:**

The proposed approach has a significant overlook for the risk of biases and untrustworthiness of LLMs. The proposed methods for medical applications pose major ethical concerns.

I strongly doubt the validity of this approach in medical applications. Specifically, what is the rationale for the generated data to be considered "correct" or incorporate the right knowledge or information? **If the LLM's output suggests a high correlation between certain marginal groups and a high prevalence of sexually transmitted diseases (STDs), how do you tell whether this is based on medical publications or social biases? Factual tracing for LLMs is currently known as a very hard problem. Active research is ongoing and there are not yet effective ways to relate the model's output to its training samples.**  Without due effort investigating this issue, using data generated by black-box models as the foundation for building medical applications is irresponsible and poses ethical concerns. Given that the work targets ultra-low resource scenarios, it is especially alerting to associate the risk with historically disadvantaged or marginal groups.

Without specific treatments, LLMs are generally quite poor in logical/mathematical reasoning. It is often challenging for these models to identify the simplest patterns in input data, which is consistent with the reported no-context generation scenario. For the generation example provided in the appendix, I found it rather concerning.

GPT models are trained overwhelmingly on web text, which contains a high level of subjective arguments, biases, and ungrounded claims especially for major social topics such as COVID or COVID patients. It could easily incorporate bias between demographic attributes and medical conditions and outcomes. The most concerning part is I don't see discussions on it at all, which makes me worry that the authors may not be fully aware of the tool they are leveraging. Given the seriousness of medical applications, this level of overlook is worrisome.

---

> ### Author Response · Authors · 2023-11-19
> **Response to Reviewer Nkcr [Part 1/5]**
>
> Dear Reviewer Nkcr,
>
> Thank you for your thoughtful comments and suggestions! We give answers to each of the following points in turn and highlight the updates to the revised manuscript. In addition, we have uploaded the revised manuscript. We hope this response alleviates your concerns, but please let us know if there are any remaining concerns.
>
> - A) Risk of biases and untrustworthiness of LLMs **[Parts 2,3/5]**
> - B) Technical contributions of CLLM **[Part 4/5]**
> - C) Answers to the other questions  **[Part 5/5]**

---

> ### Author Response · Authors · 2023-11-19
> **Response to Reviewer Nkcr [Part 2/5]**
>
> ### (A) Risk of biases and untrustworthiness of LLMs
> Thank you for raising this important concern. We agree that there are risks of bias, untrustworthiness or noisy generation when using large language models (LLMs), as explained in Sec. 1 and Sec. 2.2 in our manuscript, especially for sensitive applications like healthcare and finance. This motivated our contribution of post-generation curation as a step towards addressing this issue. That said, we agree more should be done to discuss these challenges of using LLMs in the paper.
>
> We also believe that this should not prohibit research involving LLMs in these areas. In particular, the contribution of this work is the use of curation for improved synthetic data quality. Thereby we aim to improve existing LLM-based generation methods like baseline GReaT, published at ICLR 2023, which generates synthetic data through LLMs without any curation or bias considerations. Similarly, for other generative models considered in our paper. Though CLLM does not target bias directly, let us elaborate on how it can already help *reduce* bias in baselines:
>
> 1. **CLLM can reduce discriminatory correlations.** Let us assume the LLM is biased, e.g. an LLM produces unfair correlations due to historical bias in the LLM's training data. If $D_{train}$ is unbiased, the CLLM curation step will aim to filter out this bias: the curation step analyzes each generated sample based on its predictive confidence and uncertainty with respect to a model trained on **real data** $D_{train}$, hence filters out correlations that do not fit the $D_{train}$ distribution (including discriminatory correlations in the LLM output). Specifically, Figure 5 in our manuscript shows the value of curation to align _any_ synthetic data with the correct true data $P(y|x)$, i.e. feature-label relationships. Hence, a key consideration for the practitioner is carefully construction of $D_{train}$.
> 2. **CLLM can reduce representational bias in real data.** Additionally, we show that CLLM improves the quality of synthetic data for underrepresented groups in the population. This hints at the possibility of using CLLM to remove representational bias---i.e. we could increase the $D_{train}$ with synthetic CLLM data from underrepresented groups. As we show quantitatively in Table 1 and Table 2 in our manuscript, data generated with CLLMs has a better alignment with the ground-truth distribution, compared to using a conventional generative approach (TVAE) --- already widely used in practice.
>
>
> Additionally, in terms of responsible validation, we want to highlight the "train on synthetic, test on real (TSTR)" approach adopted in our paper (see Table 3) --- a well-established approach in the synthetic data literature. This method serves as a crucial validation step for our curation process. By training a downstream model on the curated synthetic data and subsequently testing on real-world data, we establish a practical "gold-standard" benchmark to assess whether our augmented data reflects the same properties as real data. i.e. If our curation process failed to adequately filter out biased or inaccurate samples, this would be reflected in poor performance when the models are tested on real data. We show across multiple real-world datasets in Table 3 that curation greatly improves TSTR performance over a vanilla LLM and in fact mirrors real-world Oracle data performance. This implies the curation step reflects the properties of the real data.

---

> ### Author Response · Authors · 2023-11-19
> **Response to Reviewer Nkcr [Part 3/5]**
>
> **Limitations and warning readers.**
> Nonetheless, we agree that CLLM is only a first step towards better synthetic data and more can be done to build on top of it. For instance, it does not target fairness directly, e.g. if training data $D_{train}$ contains bias, this will be reflected in the curated data. Similarly, the filtering out of harmful correlations will generally not be perfect. In practice, we thus agree that output of generative models, including CLLM, should always be exhaustively evaluated on quality, fairness, and other forms of bias, before application to sensitive areas like healthcare.
>
> We have consulted an academic ethicist to discuss how these ethical concerns should be best communicated. To ensure readers may not inadvertently cause bias and harm in the real world through applying CLLM, we have included the following warning at the end of the introduction:
>
> *LLMs may make errors and may reflect or exacerbate societal biases that are present in their data [R1]. Though the curation in CLLM improves synthetic data quality, it does not directly aim to remove biases. The quality and fairness of generated data should always be evaluated. More research into LLM bias is required before methods like CLLM should be applied to real-world sensitive settings like healthcare and finance.*
>
> Furthermore, we extended our previous Ethics statement which covered consultation with stakeholders, to additionally include this warning and will include the same warning in the code repository.
>
> Thank you again for bringing up this important issue and we hope that our clarifications have addressed your concerns. If you have any other suggestions for reducing possible harm, please do let us know.
>
> **UPDATE:** we have updated Section 1 and our Ethics statement in our revised manuscript with a warning.

---

> ### Author Response · Authors · 2023-11-19
> **Response to Reviewer Nkcr [Part 4/5]**
>
> ### (B) Technical contributions of CLLM
>
> On technical contributions, we discuss the two dimensions mentioned by the reviewer: (i) Contrast to NLP augmentation and (ii) Comparison to (random) noise perturbation.
>
> **(i) Contrast to NLP augmentation**
> We would like to thank the Reviewer for bringing up the point about data augmentations in modalities such as NLP. While similar in objectives, we want to stress that data augmentation for tabular data and data augmentation in "traditional NLP" are two very different problems that have traditionally been tackled with solutions of different natures. While language models can generate coherent and contextually relevant text, tabular data augmentation is a challenging problem where the difficulty lies in preserving and extrapolating the relational dependencies and statistical characteristics within the data. This is all the more arduous as we posit for the ultra-low data regime ($n_{lab}<100$, see answer below).
>
> Furthermore, these inherent challenges entailed by tabular augmentation in the ultra-low data regime motivate the key technical contribution of CLLM which is **post-generation curation**. We demonstrate in Table 3 across 7 real-world datasets that our curation mechanism plays a key role in improving data augmentation quality and utility and is an aspect overlooked by previous works. In particular, we show that with very few samples, the augmented and curated data generated by CLLM leads to downstream performance close to that of the Oracle baseline. Furthermore, we highlight our curation mechanism tackles an overlooked aspect that not only benefits LLM tabular generation but is also applicable and beneficial to other baseline tabular generation methods.
>
>
> **(ii) Comparison to (random) noise perturbation**
> We appreciate the suggestion of the Reviewer for an experiment with a *random noise perturbation* baseline. We clarify that we considered this point in our manuscript, but realized that it has not been made clear enough. In our experiments in Sec.  3.1, we use the baseline **SMOTE** [R2], which consists in augmenting the dataset with random interpolations between samples in $D_{\mathrm{train}}$, and is thus noise injection around a given sample for data augmentation. As we show in Table 3, CLLM outperforms SMOTE by a large margin on almost all the datasets, both with and without the curation. This performance gap highlights the capacities of the LLMs to extrapolate and perform data augmentation, as we explain in Sec. 2.1, and shows that CLLM should be prefered over this baseline.

---

> ### Author Response · Authors · 2023-11-19
> **Response to Reviewer Nkcr [Part 5/5]**
>
> ### (C) Additional questions
>
>
>  * "What is the ultra-low data regime?
>
>  We apologize if this point has not been made clear enough in our submitted manuscript. We used the words "ultra-low" for settings where $\vert D_{\mathrm{train}}\vert = n_{lab} < 100$, and updated our manuscript accordingly.
>
>  We also want to emphasize that our setting differs from the typical few-shot learning literature, as we are not using the LLMs to issue predictions on given test samples. Instead, we prompt the LLM to generate new samples for data augmentation given a small $D_{\mathrm{train}}$, where the output is higher-dimensional than for few-shot prediction which only outputs labels and can typically operate with very few samples.
>
> * Diversity in generated samples & decoding strategies (temperature or sampling)
>
> We desire diversity in the generated samples, defined by the distribution that the LLM is sampling from. This is why we include the instruction of generating diverse samples in the prompt. Indeed, this information is part of the context window of the LLM and hence conditions the distribution that the LLM models, as explained in **Section 2** in our manuscript: the prompt and the in-context examples can be compactly grouped in a vector $\Phi$, defining a distribution $P_{\Phi}$.
>
> We appreciate the Reviewer's suggestion on decoding strategies. We notice that these strategies are not orthogonal to our prompting choices. Indeed, sampling assumes that a given distribution is already specified. Temperature scaling also assumes that a distribution is given, and is a way to modify its entropy in a post-hoc manner. In these two cases, we need to define an initial distribution, which we do via our prompting choices.
>
> * Clarify Table 3 ("adult row")
>
> We apologize for not making this point clearer in the manuscript. In Table 3, we bolded the best performer for each augmented dataset when the performance of a downstream model trained on that dataset was better than a model trained on $D_{\mathrm{train}}$. Hence we did not bold numbers in the "adult" row with $n=20$, as the model trained on $D_{\mathrm{train}}$ outperformed the different augmented datasets. Note, as we show in Sec. 3.2. we can also flag such poor performance proactively using our "hardness proxy".
>
>
> * Update to include standard deviation
>
> Since we have many results across many baselines, for clarity we reported average performance. However, we agree with the reviewer and have added the detailed results with standard deviations in Table 10 in our revised manuscript.
>
> **UPDATE:** We have added these results in the updated manuscript Table 10, in Appendix C.2.
>
>
> We hope this answers your points, please let us know if there are any remaining concerns.
>
>
> ----
>
> ### References
> [R1] Li, Hanzhou, John T. Moon, Saptarshi Purkayastha, Leo Anthony Celi, Hari Trivedi, and Judy W. Gichoya. "Ethics of large language models in medicine and medical research." The Lancet Digital Health 5, no. 6 (2023): e333-e335.
>
> [R2] Fernández, Alberto, et al. "SMOTE for learning from imbalanced data: progress and challenges, marking the 15-year anniversary." Journal of artificial intelligence research 61 (2018): 863-905.

---

> ### Comment · Reviewer_AZSZ · 2023-11-21
> **Clarifications**
>
> Dear Reviewer Nkcr,
>
> Thank you for your points. As I wanted to consider your review in making a decision about my score, I'd like to ask for a few clarifications.
>
> 1. Regarding your criticism of the technical contribution of this work and your point that *"this essentially comes down to the few-shot generation problem, where there is much more existing work beyond tabular data."*, would you please be able to provide a reference to an existing work that successfully solves the tabular data augmentation use case in the low-data regime that you believe would be more promising than the method proposed by the authors?
>
> 2. Thank you for pointing out the LLM bias concerns. While bias considerations are of paramount importance, this paper proposes a method for tabular data augmentation which helps improve performance on downstream tasks. In fact, the paper shows that underrepresented subpopulations in the dataset benefit the most from the proposed method. Additionally, bias and fairness assessment of models in applications is quite an established and rigorous process with many available definitions of fairness for ML models [1]. While the importance of fairness audit cannot be overstated, the responsibility for final bias and fairness assessment lies with the developers of the downstream models in applications. If a predictive model leverages the data augmentation approach proposed in this paper and passes the fairness audit conducted by its developer (or independent entity), would it be a problem to use such a model just because it leveraged CLLM for data augmentation? Would you argue that any LLM-related paper in general should then be denied publication because of the potential to be misused?
>
> Nevertheless, I would like to thank Reviewer Nkcr for bringing up the ethical considerations and I would like to commend the Authors on their excellent response and consulting an academic ethicist on communicating these ethical concerns in the ethics statement of the paper.
>
> **I would like to emphasize to the AC, Reviewer Nkcr, and Authors that I considered this review and especially the discussion on ethical concerns in my decision to recommend acceptance of the paper. In fact, the response of the authors on the ethical concerns was one of the reasons I raised my score. Regarding the criticism of the technical contribution of this work, in the absence of references supporting it, I decided to not deduct any points from my score for now.**
>
> **Question to Reviewer Nkcr:** Given the authors' responses, is 1 still a fair assessment of the quality of this work in your opinion?
>
> Thank you!
>
> References:
> [1] Mehrabi, N., Morstatter, F., Saxena, N., Lerman, K. and Galstyan, A., 2021. A survey on bias and fairness in machine learning. ACM computing surveys (CSUR), 54(6), pp.1-35.

---

> ### Comment · Reviewer_Nkcr · 2023-11-22
> **Response to authors and Reviewer AZSZ**
>
> I thank the authors for the detailed response and I personally appreciate this informative discussion. I also appreciate other high-quality reviews this work enjoyed. I would like to make more input into this constructive discussion. I apologize for my limited bandwidth. So please forgive me for not being able to write a more formal response.
>
> 1. Regarding the tabular data generation question raised by Reviewer AZSZ, no, I do not know better about how to effectively generate data in low-resource scenarios. But I think this is orthogonal to my comment on this manuscript. I included in my review that this problem is very important and timely. My concern is I do not see substantial **technical insights** offered by this work and I also have doubts about the experiment results.
>
> 2. On technical contribution, my concerns are two-fold. First, this work does not provide information gained on techniques with language models. As I mentioned in my review, I am aware of the difficulty with generating training data with LLMs in some other tasks. [1] is an example for reference. As can be seen, even for the simplest task of generating data for sentiment analysis, which might be the most natural and suitable case for LLMs, it takes substantial expertise to tune multiple aspects of the model/decoding/promoting to achieve good results. Given that inferring patterns from numerical data seems a much harder task, I would expect work contributing to this topic–even not to go further–to at least have some technical insights for the generation process. I think it will have a substantial effect on the quality of the data generation pipeline built on top of this. This work left out this part completely.
>
> [1] Tuning Language Models as Training Data Generators for Augmentation-Enhanced Few-Shot Learning
>
> For post-generation curation, the idea of incorporating this stage into the data processing pipeline is favorable. However, this data curation technique by itself isn't particularly novel compared to existing work dedicated to data evaluation/curation (e.g., [2]). To my simple understanding, the curation process is to examine the consistency of the samples with other data or the model's prediction and track through model checkpoints.
>
> [2] TRIAGE: Characterizing and auditing training data for improved regression
>
> So, my afterthoughts of reading this paper are–this is an interesting topic, this is a pioneer work. It is a bit unfortunate the authors cannot go further into the data generation process and share some insights into techniques with language models, which would be really desirable.
>
> 3. My doubt remains about the experiment results. SMOTE is different from augmentation with random noises. SMOTE is often not very effective in populating insufficient data and often leads to worsening performance. So I am not surprised SMOTE performance is somewhat lower–it is often the case. **My concern is most of the baselines listed here achieve a worse performance than D_train–without any generation/curation at all.** So I wonder if these are valid baselines or results. At the same time, the performance improvement from CLLM is marginal. I think with careful augmentation of D_train and proper curation, it is possible to match the performance of CLLM. This goes back to my point in the beginning–I do not see the technical insights with LLM I think are needed for the generation process to succeed (in a meaningful way).
>
> Conditioning on my prior doubt about the factuality of LM-generated contents (I worked on fact tracing/attribution for generative models),  I don't have enough confidence that the data is generated in the way claimed in this paper.
>
> As much as I respect the hard work of the authors, I don't have the evidence I need to support this work.
>
> 4. My previous score of 1 is for the lack of discussion on ethical concerns, which I perceive as a critical issue. Now this has been appropriately resolved, I raise my score to 5.
>
> I am not passionate about supporting this work for publication at its current stage and I am not passionately against it, either. This is a decent work and I appreciate the authors' and reviewers' hard work and dedication. My honest opinion is I would like to see work on language technology to go beyond intuitive prompting.

---

> ### Author Response · Authors · 2023-11-22
> **Thank you for your response and score increase [Part 1/2]**
>
> Dear ``Reviewer Nkcr``,
>
>
>
> Thank you for engaging with us and for your time in the review process! We are glad to have addressed your concerns about the ethical considerations and want to thank you for helping us improve the paper on this important aspect.
>
>
> In this response we aim to achieve two things: to answer the points in your last response, but also to make the case for you to consider moving from a 5 to a 6 🙂.
>
> Thank you for your honesty in saying you are _"not passionately against [publication]"_. We would like to try and convince you to favour publication (even if not passionately 😉). We agree with you that this topic is _"highly relevant and [...] particularly helpful for disadvantaged or marginal groups or rare scenarios"_, and that the _"research need is valid and important"_ due to it being understudied by the ML community. We thank you for acknowledging it as an _"interesting topic"_ and a _"pioneer work"_, yet we acknowledge that technical improvements are possible. **We are optimistic that the publication and simplicity of CLLM will spur further debate and research, including into improving prompts and other curation mechanisms** (e.g. for algorithmic fairness or privacy). We hope this alone makes it a paper worthwhile publishing.
>
>
>
>
> Let us respond to the different points you raised in your last comment.
>
> ----
>
>
> ### Using LLMs requires expertise and tuning, and [1]
>
>
>
> We agree that the use of LLMs is not straightforward, that tuning and prompting can have a large influence on the output quality, and that we should have highlighted this better in the paper. Though we find different techniques for tuning and prompting LLMs interesting, it is beyond the focus of our work---i.e. improving LLM synthetic data through data curation. Despite the difficulties of LLM prompting and tuning, we hope to have shown that CLLM's simple prompting approach compares very favourably to existing tabular generative models. Some of the insights our paper _does_ provide with respect to LLM generation are listed below:
>
> (1) **Importance of context:** Figure 4 and Table 2 show that CLLM leverages the contextual information of the in-context samples, which motivates our prompt design.
> (2) **Extrapolation capacities of the LLM:** Figure 2 and Figure 8 show the capacity of the LLM to extrapolate beyond the limited in-context samples.
> (3) **Feature-label relationship is captured accurately by the LLM:** Figure 5 shows that the data generated by the LLM captures the feature-label relationship better than the baselines.
>
> **To clarify to readers that LLMs are not an easy silver bullet, we will add the following to the Discussion's new limitations paragraph:**
>
> >_"We have shown how CLLM improves data augmentation through curating the LLM. Nonetheless, further improvements may be achieved through different tuning and prompting of the LLM, as evidenced in different domains (e.g. see [1] and (Liu et al, 2023)). Improving LLM tuning and prompting is beyond the scope of our work, but we regard this as a promising avenue for future work."_
>
> ----
>
> ### Curation contribution and comparison to [2]
>
>
>
> We agree that data curation is a general idea in ML and thank the reviewer for referencing the related work [2]. We agree both approaches draw inspiration from learning theory and leverage the idea of learning dynamics via checkpoints. Otherwise, we would like to argue that [2] and our paper are quite distinct in contribution:
>
> (1) **Problem and aim**: We are (to the best of our knowledge) the first to use curation for data augmentation, whereas [2] focuses on real data selection focussed on improving regressors. Additionally, we compute the training dynamics of $D_{syn}$ with respect to a model trained on a small gold-standard $D_{train}$ to assess quality and remove inconsistent samples, whereas [2] aims to directly operate on and audit $D_{train}$---discarding _real_ samples that it thinks are inaccurate.
> (2) **Types of samples flagged**: We flag samples that have the incorrect $P(y|x)$ with respect to a real dataset, whereas [2] detects samples that would be systematically over or under-estimated by a regressor.
> (3) **Different curation metrics:** We use the confidence and aleatoric uncertainty computed on the training dynamics, whereas the authors of [2] compute a conformal predictive distribution for a regressor---hence is not applicable to our setting.
>
> As an additional point, we wish to highlight that **[2] has only become available online on October 29th after the ICLR submission deadline.** Regardless, we will include it in our related work.
>
> To summarize, we wish to highlight that the idea of curation for tabular data *augmentation* has not been explored prior to this work and we hope CLLM will open up further exploration in this area.
>
> ``response continues``

---

> ### Author Response · Authors · 2023-11-22
> **Thank you for your response and score increase [Part 2/2]**
>
> ### Choice of baselines and random noise
>
> **(i) Choice of baselines:** We first clarify our existing baselines, which encompass a broad and representative spectrum of different classes of generative modeling techniques used for tabular data augmentation (Qian et al, 2023): GAN, VAE, Normalizing Flow, LLM, and Score-based modeling, as well as classical augmentation (e.g. SMOTE). As the Reviewer rightfully mentions, the baselines struggle in the ultra-low data regime. This illustrates a key research gap: traditional data augmentation approaches struggle in such ultra-low sample settings, which CLLM addresses with the LLM and its generative capabilities.
>
> **(ii) Random noise baseline:** We thank the Reviewer for the suggestion of the random noise baseline. We performed a new experiment, where we augment the dataset with random additive Gaussian noise. In order to capture the correlations between the different features, we fit a Kernel Density Estimator with a Gaussian kernel and bandwidth given by Scott's rule. We then sample $1000$ points to create an augmented dataset $D_{\mathrm{syn}}$.
>
> We report the performance gap between CLLM and this baseline (with and without curation) for the Covid and Compas datasets. The result can be found at the following link: **https://i.imgur.com/iUkTZ5w.png**
>
>
> We observe that the random noise baseline does not match the performance of CLLM, although the baseline naturally improves as the dataset $D_{\mathrm{train}}$ grows in size.
>
> **UPDATE**: We include this result in a new Appendix C.5 in our revised manuscript.
>
> ---
>
> _We thank you for your time and energy in the reviewing process, please let us know if you have any additional questions._
>
> ---
>
> ### References
> Liu, P., Yuan, W., Fu, J., Jiang, Z., Hayashi, H., & Neubig, G. (2023). Pre-train, prompt, and predict: A systematic survey of prompting methods in natural language processing. ACM Computing Surveys, 55(9), 1-35.
>
> Qian, Zhaozhi, Bogdan-Constantin Cebere, and Mihaela van der Schaar. "Synthcity: facilitating innovative use cases of synthetic data in different data modalities." arXiv preprint arXiv:2301.07573 (2023).

---

> ### Comment · Reviewer_Nkcr · 2023-11-23
>
> Thanks to the authors for the response and additional results. I have read this in full. I still have to maintain my standing on this work. As much as I appreciate the authors for their dedication to this work, which is inspiring to me, I cannot go further with my review without the evidence I need to establish scientific confidence.
>
> I am not as familiar with the state-of-the-art performance of tabular data generation methods in this specific "ultra-low data regime". I remain a scientific doubt about this result. I decrease the confidence of my review by 1 and leave the evaluation of this part to other reviewers and the AC who have this expertise.

---

> > ### Author Response · Authors · 2023-11-23
> >
> > Thank you for your response, and thank you again for your engagement and feedback. We will hope for the best🙂
> >
> > Authors of Submission #7282

---

### Official Review · Reviewer_AZSZ · 2023-11-01

**Soundness:** 3 good
**Presentation:** 4 excellent
**Contribution:** 3 good
**Rating:** 8
**Confidence:** 5

**Summary:**

The paper proposes to use strong LLMs such as GPT-4 for tabular data augmentation and generation. The paper also proposes a generated data curation technique based on predictive confidence and aleatoric uncertainty metrics. The paper shows that the resulting method CLLM is able to effectively leverage prior knowledge of LLMs and achieves good experimental performance in low-data regimes.

**Strengths:**

Strengths:
* The paper proposes an intuitive and well-motivated approach to leverage background knowledge of GPT-4 for informed tabular data generation
* The experimental results are promising and cover a representative set of baseline tabular data generation methods
* The idea of generated data curation is very interesting. Discarding samples with both low predictive confidence and data uncertainty improves performance both of the GPT-4-based data generation and of other tabular data generation model
* The additional insight into GPT-4’s ability to extrapolate to unseen regions of the manifold is interesting, even though the evidence is somewhat anecdotal and based on a TSNE plots.
* Importantly, the paper shows that LLM-based tabular data generation is most helpful for generation of underrepresented samples which traditionally is a challenge for other tabular data generation methods.
* The presentation is excellent.

**Weaknesses:**

Weaknesses and Questions:
* The paper only tests generation with GPT-4. It would be interesting to see if the findings in the paper are consistent across LLMs, for example, including Claude 2 or LLAMAv2 would be helpful.
* Could you provide an explanation for GPT-4 generated data outperforming D_oracle in the Train-on-synthetic-Test-on-real setting in Table 3?
* What is your dataset selection logic? It would be helpful to include more datasets from the papers of other tabular data generation baselines.
* Do the performance gains come solely from the background knowledge of GPT-4 or is GPT-4 also able to build a strong data model? Have you tested CLLM on any datasets where the background knowledge of GPT-4 would not be as useful, for example, on datasets with anonymized features? Such an experiment would help further understand the source of performance gains.
* Could you explain the utility drop for GPT-4-no-context on 200 samples in Table 2? Why would more data lead to degraded performance? Could this be simply caused by randomness in the results?
* If the results in Table 2 are indeed volatile, it would be very helpful to include error bars and bold all statistically significant winners. The error bars could be constructed by, for example, running the simulation for multiple seeds or resampling D_train. Right now, for example Recall of 0.89 is bolded for GPT-4 w/context for 40 samples and not bolded for GPT-no-context for 40 samples. This makes the results in Table 2 unconvincing in their current form. Although it is reasonable to hypothesize that including the background information about a dataset is helpful, as I mentioned above, a better validation of that would be helpful.
* Related, which dataset are the results in Table 2 based on? Are there similar results for other datasets? What about the dataset behind Figure 3 and Table 1?
* Related work is currently limited and would benefit from including other related papers. While a few examples of tabular data generation methods are included and used as baselines, at least citing other prominent tabular data generation methods such as STaSy[1] would be helpful. Additionally, even though the introduction mentions that the low-data problem is undervalued, a few tabular deep learning works in fact address extreme low-data regimes, some of them also include experiments in the medical domain [2,3,4,5]. For example, [2,3] are tabular transfer learning approaches with experiments in extreme low-data regimes in the medical domain, while [4] and [5] are knowledge-graph-augmented tabular approaches enabling performance improvements in low-data regimes of wide and short datasets. The idea of leveraging a knowledge graph is similar to the idea of using an LLM in that they both rely on prior knowledge. It would definitely be helpful to cite these works, including them in experiments might be tricky because of the knowledge graph construction.


References:

[1] STaSy: Score-based Tabular data Synthesis, ICLR 2023 (https://openreview.net/forum?id=1mNssCWt_v)

[2] Levin, R., Cherepanova, V., Schwarzschild, A., Bansal, A., Bruss, C.B., Goldstein, T., Wilson, A.G. and Goldblum, M., 2022. Transfer learning with deep tabular models. arXiv preprint arXiv:2206.15306.

[3] Benchmarking Tabular Representation Models in Transfer Learning Settings, NeurIPS 2023 Tabular Representation Learning Workshop (https://openreview.net/forum?id=HtdZSf1ObU)

[4] Margeloiu, A., Simidjievski, N., Lio, P. and Jamnik, M., 2022. Graph-Conditioned MLP for High-Dimensional Tabular Biomedical Data. arXiv preprint arXiv:2211.06302.

[5] Ruiz, C., Ren, H., Huang, K. and Leskovec, J., 2023. Enabling tabular deep learning when $ d\gg n $ with an auxiliary knowledge graph. arXiv preprint arXiv:2306.04766.

**Questions:**

Please, see the weaknesses section for both weaknesses and questions.

**Details Of Ethics Concerns:**

no ethics concerns

---

> ### Author Response · Authors · 2023-11-19
> **Response to Reviewer AZSZ [Part 1/4]**
>
> Dear Reviewer AZSZ,
>
> Thank you for your thoughtful comments and suggestions! We give answers to each of the following points in turn and highlight the updates to the revised manuscript. In addition, we have uploaded the revised manuscript. We hope this response alleviates your concerns, but please let us know if there are any remaining concerns.
>
> - A) Background knowledge of LLMs **[Part 2/4]**
> - B) Related work **[Part 2/4]**
> - C) LLM backbone **[Part 3/4]**
> - D) Answers to the other questions **[Parts 3,4/4]**

---

> ### Author Response · Authors · 2023-11-19
> **Response to Reviewer AZSZ [Part 2/4]**
>
> ### (A) Background knowledge of LLMs
>
> We would like to thank the Reviewer for the astute point on disentangling the background knowledge of LLMs. As the Reviewer rightfully points out, two components can be attributed to the good performances of CLLMs: the background knowledge of the LLM, and its capacity to build a strong data model. Additionally, the Reviewer asks how would anonymized features affect the results.
>
> Let us tackle in what follows (i) anonymized features and (ii) background knowledge.
>
> (i) **Anonymized features:** Firstly, in terms of anonymized features, let us clarify that our ablation study in **Section 2.1 (page 5)** tackles this point. We assess the importance of contextual information in the prompt, where we test a variant called *prompt w/ no context* which anonymizes the features as feat1, feat2 etc and only provides the numerical in-context examples, without explanation of what these features are. The results in Table 2 (on the Covid dataset) show that the lack of feature information reduces precision and recall and also reduces the performance of a downstream model (utility). This shows the importance of providing contextual information to exploit the potential of the LLM, and hence motivates the prompt that we use in our work.
>
> We have also run an additional experiment on the Compas dataset, similarly ablating with anonymized features and similarly demonstrating the importance of contextual information in the prompt itself. We report the results in our response below (in **D) Answers to the other questions**) in reference to one of the other Reviewer's questions.
>
> **UPDATE**: We have updated Appendix C.4 to show the result on the Compas dataset.
>
> (ii) **Background information vs ICL to build a data model**:
> We followed the suggestion of the Reviewer and conducted a new experiment to understand the effect of the LLMs background knowledge (e.g. prior).
>
> We considered the Covid dataset (private medical dataset, to avoid memorization issues) and generated data with GPT-4 (same as in Sec 2.1). We ablate the prompt used in our work (detailed in Appendix B.5), and solely provide one in-context example in the prompt, in order to give the LLM the *minimal* amount of information about the desired structure of the dataset.
>
> This lack of examples forces the LLM to rely on its own prior (background knowledge), and removes the effect of in-context examples which could be used to build a data model.
>
> We report the results for CLLM in the following table:
> | In-context samples |  Downstream accuracy  |
> |--------------------|----------------|
> | n=1 (Prior)        | 70.20+-1.60    |
> | n=20               | 73.87+-0.50    |
> | n=40               | 73.95+-0.67    |
> | n=100              | 74.71+-0.34    |
> | $D_{\mathrm{oracle}}$| 74.6 +- 0.15 |
>
>
>
> From these results, we conclude the following:
>
> (1) The LLM prior permits to obtain good downstream performance, but it is outperformed by $\mathcal{D_{oracle}}$ by a margin of $4.4\%$. Hence, we cannot solely rely on the prior.
>
> (2) Downstream performance increases as the number of in-context samples increases. This shows it is indeed important to include the in-context examples if we wish to obtain downstream performance close to $\mathcal{D_{oracle}}$, as the LLM can build a good data model.
>
> This implies that while the LLM does use background knowledge of similar datasets, it still requires in-context samples to refine its prior by creating a good data model.
>
> **UPDATE**: We included these results in Appendix C.3 of our updated manuscript
>
> We thank the reviewer for their suggestion, where this examination of different aspects of the LLM's background knowledge has strengthened the paper.
>
> ----
>
> ### (B) Related work
>
>
> We would like to thank the Reviewer for pointing these five references out. We agree with the Reviewer that using a knowledge graph is a relevant way to inject prior knowledge in the low-data regime, even though constructing them is not always straightforward. As such, we thank the Reviewer for rightfully mentioning that the LLM could serve a similar purpose (while being flexible with regard to prior knowledge).
>
> **UPDATE**: We have included the references in our Related Work section as well as covered them further in the Extended related work section in Appendix A.

---

> ### Author Response · Authors · 2023-11-19
> **Response to Reviewer AZSZ [Part 3/4]**
>
> ### (C) LLM backbone
> We acknowledge the reviewer's suggestion to test our approach with other large language models such as Claude 2 or LLAMA2.  We agree that exploring the consistency of our findings across various LLMs such as Claude and LLAMA2 is an interesting avenue for future research. We want to highlight that CLLM aims to underscore the broader applicability of using an LLM with post-generation curation for data augmentation and used both GPT 3.5 and GPT-4 as examples. Extending our experiments to include Claude 2 or LLAMA2 could indeed provide additional insights, though we want to clarify that our paper's primary contribution lies in demonstrating and validating the general concept of leveraging LLMs for data augmentation in ultra-low data regimes, as well as highlighting the value of our post-generation curation mechanism of the LLM output, rather than conducting a comparative analysis between different LLMs.
>
> ----
>
>
> ### (D) Answers to the other questions
>
> * Why does GPT-4 sometimes outperform $\mathcal{D}_{oracle}$ in TSTR?
>
> As rightfully noted by the Reviewer, GPT-4 outperforms $D_{\mathrm{oracle}}$ on the dataset Compas for some settings, in Table 3. We attribute this phenomenon to the inherent noisiness (e.g. label noise) of this dataset, as we also notice that this dataset leads to the worst downstream accuracy out of the 7 datasets in Table 3, on $D_{oracle}$.
>
> * What is your dataset selection logic?
>
> We wish to clarify our selection of the 7 datasets we use --- both open-source and private datasets.
>
> (1) **Open-source:** Adult, Drug and Compas are widely used open-source datasets used in the tabular data literature. Adult and Drug are both UCI datasets that have been used in many papers, while Compas is part of OpenML [R1]. Our reason for selecting them is that, despite them being open-source, they are highly reflective of domains in which we might be unable to collect many samples --- hence in reality would often be in an ultra-low data regime.
>
> (2) **Private datasets:** We wanted to disentangle the possible role of memorization in the strong performance of the LLM. To ensure the datasets are not in the LLMs training corpus, we selected 4 private medical datasets that need an authorization process to access. Hence, these datasets would not be part of the LLMs training corpus given their proprietary nature and hence would be unseen to the LLM. While the private and unseen aspect was the main motivation, we also wish to highlight that these are real-world medical datasets. Consequently, this allows us to test a highly realistic problem setting.
>
> We apologize if this was unclear and have attempted to better clarify this in our description of dataset selection in Appendix B.1.
>
> **UPDATE**: We have added this discussion in Appendix B.1 of our revised manuscript.

---

> ### Author Response · Authors · 2023-11-19
> **Response to Reviewer AZSZ [Part 4/4]**
>
> * Error bars on Table 2
>
>
>
> We thank the reviewer for the suggestion and have reported the results with error bars across multiple seeds with the data resampled. Please see the table below, which shows results for the Covid dataset.
>
>
> | $n_{\mathrm{samples}}$ in $\mathcal{D_{train}}$ | GPT-4 w/ context Precision | GPT-4 w/ context Recall | GPT-4 w/ context Utility | GPT-4 no context Precision | GPT-4 no context Recall | GPT-4 no context Utility | TVAE Precision | TVAE Recall | TVAE Utility |
> | ---------------------------------- | ------------------- | ------------------- | ------------------ | ------------------ | ------------------ | ------------------ | ------ | ------ | ------ |
> | 10                                 | **0.41+-0.04**      | **0.87+-0.03**      | **0.74+-0.01**     | 0.13+-0.0          | 0.82+-0.01         | 0.66+-0.01         | 0.33+-0.07 | 0.50+-0.03 | 0.59+-0.02 |
> | 20                                 | **0.40+-0.01**      | **0.91+-0.01**      | **0.76+-0.0**      | 0.11+-0.0          | 0.89+-0.0          | 0.69+-0.0          | 0.27+-0.01 | 0.68+-0.01 | 0.62+-0.03 |
> | 50                                 | **0.42+-0.01**      | 0.86+-0.02          | **0.75+-0.01**     | 0.11+-0.01         | **0.90+-0.01**     | 0.74+-0.01         | 0.39+-0.02 | 0.67+-0.03 | 0.64+-0.06 |
> | 100                                | 0.44+-0.02          | 0.85+-0.02          | **0.75+-0.0**      | 0.08+-0.01         | **0.90+-0.0**      | 0.60+-0.01         | **0.47+-0.0** | 0.73+-0.01 | 0.65+-0.02 |
>
> **UPDATE:** We have updated Table 2 in the revised manuscript with the standard deviations.
>
>
>
>
>
> * Clarifying the dataset for Table 2
>
> The results in Table 2 are for the dataset Covid. This is the same dataset that we used for Table 1 and Figure 3. We apologize if the mention of Covid as a running example didn't make this clear enough. Hence, we have updated individual captions in the manuscript to clarify this point.
>
> **UPDATE**: We have updated the caption of Table 2 in the updated manuscript to make this clear.
>
> * Additional dataset for an experiment in Table 2 (Ablation of contextual information)
>
> As suggested we have also included a result for an additional dataset (Compas dataset) to illustrate to ablate the context. The results similarly show the value of the contextual information in the prompt.
>
> | $n_{\mathrm{samples}}$ in $\mathcal{D_{train}}$ | GPT-4 w/ context Precision | GPT-4 w/ context Recall | GPT-4 w/ context Utility | GPT-4 no context Precision | GPT-4 no context Recall | GPT-4 no context Utility | TVAE Precision | TVAE Recall | TVAE Utility |
> | ---------------------------------- | ------------------- | ------------------- | ------------------ | ------------------ | ------------------ | ------------------ | ------ | ------ | ------ |
> | 10                                 | **0.69**+-0.02      | **0.88**+-0.02      | **0.69**+-0.02     | 0.27+-0.03         | **0.89**+-0.03     | 0.60+-0.03        | 0.43+-0.02 | 0.43+-0.05 | 0.55+-0.04 |
> | 20                                 | **0.70**+-0.0       | **0.92**+-0.01      | **0.65**+-0.03     | 0.31+-0.06         | 0.84+-0.03         | 0.57+-0.01        | 0.54+-0.02 | 0.80+-0.02 | 0.50+-0.04 |
> | 50                                 | **0.69**+-0.02      | **0.89**+-0.02      | **0.69**+-0.01     | 0.34+-0.1          | 0.85+-0.05         | 0.62+-0.01        | 0.60+-0.03 | 0.86+-0.02 | 0.59+-0.03 |
> | 100                                | **0.70**+-0.01      | **0.89**+-0.02      | **0.69**+-0.01     | 0.31+-0.05         | 0.87+-0.03         | 0.58+-0.05        | 0.65+-0.02 | 0.88+-0.01 | 0.63+-0.01 |
>
>
> **UPDATE**: We have included this result in a new Appendix C.4 in our revised manuscript.
>
>
> We hope this answers your points, please let us know if there are any remaining concerns.
>
>
> ----
>
> ## References
> [R1] Joaquin Vanschoren, Jan N. van Rijn, Bernd Bischl, and Luis Torgo. OpenML: networked science in machine learning. SIGKDD Explorations 15(2), pp 49-60, 2013.

---

> > ### Comment · Reviewer_AZSZ · 2023-11-21
> > **Thank you for the thorough response**
> >
> > Dear Authors,
> >
> > Thank you very much for your thorough response, additional experiments, ablations, and clarifications. Your experiments disentangling the helpfulness of the background LLM knowledge and ICL provide valuable insight for anyone looking to apply this approach in practice. Thank you for the explanation about D-oracle performance on Compas -- it makes sense. Also, thank you for clarifying your dataset selection logic -- you have a great point about using non-open data to alleviate memorization concerns, please mention it in the main body of the paper.
> >
> > Given the cost of GPT-4 and the API access limiting control of reproducibility of its outputs, I still believe it would be interesting to see if smaller open-source LLMs can be leveraged for the data augmentation purpose and how far their performance would be from GPT-4 because it may be beneficial for users of your framework to have complete control over the model and data (for privacy, security and other reasons). However, I agree that there is only marginal value in running Claude experiments and I welcome your point that the paper's primary contribution lies in demonstrating and validating the general concept of leveraging LLMs for data augmentation in ultra-low data regimes (and demonstrating the value of curation) rather than conducting a comparative analysis between different LLMs for this purpose. I'd like to suggest this for your consideration as a direction for your future work and/or your code base on GitHub as this may significantly increase the impact of your work for practitioners.
> >
> > Thank you for providing the error bars for the tables I requested and for providing the error bars for Table 3 in Table 10 of the Appendix. I trust that between the main body and appendix you have now included error bars for all reported results and comparisons of methods.
> >
> > With that, I am happy to strongly recommend acceptance and increase my score to 8.

---

> > > ### Author Response · Authors · 2023-11-21
> > > **Thank you for your engagement**
> > >
> > > Dear Reviewer,
> > >
> > > Thank you for your feedback and engagement. Thank you especially for your public support and vote of confidence towards the other reviewers, this is very much appreciated.
> > >
> > > We will ensure the dataset selection reasoning will be included in the main Experiments section more clearly, i.e. how datasets are chosen to avoid possible LLM memorization biasing our results. Additionally, we will be adding a paragraph to the discussion on the limitations of LLMs and recommending future work:
> > >
> > > _**Limitations**. In this work, we studied GPT-3.5 and GPT-4 as backbones for CLLM. The cost of GPT-4 and OpenAI's API-only access pose limitations, e.g. on wide accessibility, on knowing which data was used for training the models, and on understanding the LLM's output better. Using smaller and open LLMs could overcome these limitations, though this could come with a reduction in performance. We leave this as a promising direction for future work._
> > >
> > > Thank you again for your time and feedback.

---

### Author Response · Authors · 2023-11-19
**Response Overview**

We thank the Reviewers for their insightful and positive feedback, and their time during the review process!

We are encouraged the reviewers found the ultra-low data augmentation problem interesting (``R-NKcr``, ``R-4AgE``) and “highly relevant” (``R-NKcr``) with “far-reaching consequences” (``R-4AgE``) to be "helpful for disadvantaged or marginal groups or rare scenarios"(``R-NKcr``). The reviewers also deemed our post-generation curation as “interesting” (``R-AZSZ``), “smart"(``R-NKcr``) and “relevant and timely"(``R-NKcr``). Finally, we are glad the reviewers felt “the experimental results are promising and cover a representative set of baseline tabular data generation methods" (``R-AZSZ``) and that the "insight into GPT-4’s ability to extrapolate to unseen regions of the manifold is interesting” (``R-AZSZ``), coupled with its ability to help generation of underrepresented samples addresses a problem which "traditionally is a challenge for other tabular data generation methods" (``R-AZSZ``).

We address specific questions and comments below, along with corresponding updates to the uploaded manuscript, with changes marked in purple.

On the basis of our clarifications and updates, we hope we have addressed the Reviewers' concerns.

Thank you for your kind consideration!

Paper 7282 Authors

---

### Meta-Review · Area_Chair_5oh5 · 2023-12-10

**Metareview:**

The paper uses LLMs to create synthetic data to enable learning predictive models in low-data regimes (e.g. rare diseases).  The idea is to first construct an LLM prompt with background description of text, (training) examples as demonstrations of features and labels and instruct the model to generate synthetic samples. To subselect synthetic samples that are of increased utility for a model to have better downstream generalization, the paper proposes looking at the learning dynamics of a classifier trained on the original training set. Specifically they look at the average confidence of the classifier trained on the original data at various training checkpoints. The paper is well written, formatted and easy to understand. I also think the idea itself is new and underexplored. It is also interesting that improvements are most salient on smaller subgroups.

The paper generated quite a bit of discussion among the reviewers and authors. One reviewer raised the concern about the ethical implications of the LLMs itself which prompted the author to include a warning template within the code. Another was around disentangling how much background knowledge was being used by the LLM. To this end, the authors incorporated a few newer results studying how well incontext learning does on the tasks herein.

This came down to a borderline decision -- beyond the concerns raised in the review process, its a little concerning that only closed source LLMs were used to evaluate this work; while they no doubt have the best performance and do add value to an academic discussion it is unclear what the shelf life of such work that only uses such models actually is particularly in light of the quality of such models degrading over time (https://arxiv.org/abs/2307.09009) and the easy availability of LLAMA2 chat (that can be run on a CPU via llama.cpp). Second, while much of the discussion centered around the use and utility of LLMs, very little discussion is made in the main text around the mechanism used to select the data that is then used to train the LLM. While the heuristic herein works, there is little to no discussion on what motivates it and when/why it should. This is important since in the ultra-low data regime one has a very noise estimate of the uncertainty itself (since the variance of the estimator correspond to each checkpoint can be high). Third, beyond the prior work suggested to improve the related work section of this paper, there is prior work that represents an important baseline to compare against: e.g. TabLLM: Few-shot Classification of Tabular Data with Large Language
Models, Hegelsman et. al https://arxiv.org/abs/2210.10723 also tackles the problem of improving prediction in tabular data (albeit using a very different approach) and showcase improved sample efficiency. This would be the most natural point of comparison to published related work.

Overall, I do think the approach itself is novel and encourage the authors to continue building upon the submission.

**Justification For Why Not Higher Score:**

I have provided a justification in the main review.

**Justification For Why Not Lower Score:**

N/A

---

### Decision · Program_Chairs · 2024-01-16

Reject